From morphology to molecules: a combined source approach to untangle the taxonomy of Clessinia (Gastropoda, Odontostomidae), endemic land snails from the Dry Chaco ecoregion

Cuezzo Maria Gabriela 1 gcuezzo@webmail.unt.edu.ar
Miranda Maria Jose 1
http://orcid.org/0000-0001-9660-552X Vogler Roberto Eugenio 2
http://orcid.org/0000-0002-0052-6710 Beltramino Ariel Anibal 2 beltraminoariel@hotmail.com
1 Facultad de Ciencias Naturales, Instituto de Biodiversidad Neotropical (IBN), Consejo Nacional de Investigaciones Científicas y Técnicas—Universidad Nacional de Tucumán , San Miguel de Tucumán, Tucumán , Argentina
2 Facultad de Ciencias Exactas, Químicas y Naturales, Instituto de Biología Subtropical (IBS), Consejo Nacional de Investigaciones Científicas y Técnicas—Universidad Nacional de Misiones , Posadas, Misiones , Argentina
Bieler Rudiger
Electronic publication date: 2018 Dec 6
Publication date: 2018
Volume: 6
Electronic Location ID: e5986
Received 2018 Jun 5; Accepted 2018 Oct 23
Copyright: © 2018 Cuezzo et al.
Copyright year: 2018
Copyright holder: Cuezzo et al.
License: This is an open access article distributed under the terms of the Creative Commons Attribution License, which permits unrestricted use, distribution, reproduction and adaptation in any medium and for any purpose provided that it is properly attributed. For attribution, the original author(s), title, publication source (PeerJ) and either DOI or URL of the article must be cited.
License URL: https://creativecommons.org/licenses/by/4.0/

Keywords: Stylommatophora, Spixia, Argentina, Molecular analyses, Periostracum

Funding: Argentine National Council of Scientific Research PIP 0050 (Maria Gabriela Cuezzo) and institutional, P-UE 0099 (Instituto de Biodiversidad Neotropical) Facultad de Ciencias Exactas, Químicas y Naturales, Universidad Nacional de Misiones Proyecto de Investigación 16Q634 This study was financially supported by the Argentine National Council of Scientific Research through grants PIP 0050 awarded to Maria Gabriela Cuezzo and institutional grant P-UE 0099 awarded to Instituto de Biodiversidad Neotropical and by Facultad de Ciencias Exactas, Químicas y Naturales, Universidad Nacional de Misiones (Proyecto de Investigación 16Q634). The funders had no role in study design, data collection and analysis, decision to publish, or preparation of the manuscript.

==============================
Background

Land gastropods of the Dry Chaco merit special attention because they comprise a highly diverse but barely studied group. Clessinia Doering, 1875 are typical inhabitants of this ecoregion. The inclusion of their distribution areas into Spixia range, their shell shape similarities, and a former molecular study raised doubts on the monophyly of this genus. The present study review the species of Clessinia, under a morphological, geometric morphometrics, and molecular combined approach.

Methods

Adults were collected, photographed, measured, and dissected for anatomical studies. Shell ultrastructure was studied with scanning electron microscope. Geometric morphometric analyses on shells were performed testing if they gave complementary information to anatomy. Two mitochondrial genes, and a nuclear region were studied. Phylogenetic reconstructions to explore the relationships of DNA sequences here obtained to those of Clessinia and Spixia species from GenBank were performed.

Results

Species description on shell, periostracal ornamentation and anatomy is provided. We raised former Clessinia cordovana striata to species rank, naming it as Clessinia tulumbensis sp. nov. The periostracum, consisting of hairs and lamellae, has taxonomic importance for species identification. Shell morphometric analyses, inner sculpture of penis and proportion of the epiphallus and penis, were useful tools to species identification. Nuclear markers do not exhibit enough genetic variation to determine species relationships. Based on the mitochondrial markers, genetic distances among Clessinia species were greater than 10%, and while C. cordovana, C. nattkemperi, and C. pagoda were recognized as distinct evolutionary genetic species, the distinction between C. stelzneri and C. tulumbensis sp. nov. was not evident. Clessinia and Spixia were paraphyletic in the molecular phylogenetic analyses. Species of Clessinia here treated have narrow distributional areas and are endemic to the Chaco Serrano subecoregion, restricted to small patches within the Dry Chaco. Clessinia and Spixia are synonymous, and the valid name of the taxon should be Clessinia Doering, 1875 which has priority over Spixia Pilsbry & Vanatta, 1894.

Discussion

Our results support the composition of C. cordovana complex by three species, C. cordovana, C. stelzneri, and C. tulumbensis sp. nov. The low genetic divergence between C. stelzneri and C. tulumbensis sp. nov. suggests that they have evolved relatively recently. The former Spixia and Clessinia are externally distinguished because Clessinia has a detached aperture from the body whorl forming a cornet, periostracal microsculpture extended over dorsal portion of the peristome, five inner teeth on the shell aperture instead of three–four found in Spixia. Morphological similarities exists between both genera in shell shape, type of periostracum microsculpture, reproductive anatomy, besides the overlap in geographic ranges.

Introduction

Taxonomy is a crucial discipline in biology if practiced within an evolutionary framework (Dubois, 2017). The taxonomic and biodiversity crisis requires a strong acceleration of the work of exploration, study, description, and naming of the species of the globe (Wheeler, Raven & Wilson, 2004; Dubois, 2007, 2010). However, there is a tendency towards a strong decrease in morphological and anatomical studies while “replacing” them with molecular analyses which are unable, if are used alone, to provide the wealth of diverse information on organisms which morphology, anatomy, and other biological studies offer (Dubois, 2017). Species are hypotheses, and as such it is required that they make predictions (that more data of approximately the same quality will support such groupings) and are thereby testable (that more data of approximately the same quality do not suggest alternative groupings) (Wheeler, 2004; Valdecasas, Williams & Wheeler, 2008). Then, identification of species utilized in a study impacts all subsequent comparisons or any further studies on species-specific traits or attributes.

The combination of morphological and ecological information with different molecular markers can be a good method of species identification, because it can provide an accurate perspective on evolutionary history of an organism and its taxonomic relationships (Davison, Blackie & Scothern, 2009). In this way, new methods do not replace, but complement the traditional, tested methods, and procedures (Wheeler, Raven & Wilson, 2004). Geometric morphometrics is a useful tool to accurately analyze shell variability decomposing shell form into size and shape in each species (Carvajal-Rodriguez, Conde-Padin & Rolan-Alvarez, 2005; Cruz, Pante & Rohlf, 2012; Greve et al., 2012). When multiple sources are used for analyzing a taxonomic problem, agreement among them is expected, but differences between them can also be rich revealing different aspects of a same problem and contributing to interpretation of the evolutionary patterns. Conflicts between different analyses can stimulate a new more detailed investigation of the characters and taxa involved.

The purpose and framework of our work is to study and identify endemic species of the Dry Chaco ecoregion in Argentina (sensu Olson et al., 2001) as a first step to reevaluate its taxonomic information and conservation status. The Dry Chaco is an ecoregion that merits special attention from biodiversity studies because it represents the largest continuous dry forest remnant in South America. In the past decades it ranked second in terms of deforestation after the Amazonian rainforest, mostly due to the expansion of soybean crops and planted pastures (Gasparri & Grau, 2009). Although this area is suspected to host a rich gastropod fauna, there are no current formal studies focusing on the diversity of molluscan taxa in the area.

Odontostomidae is a species-rich family of pulmonate snails distributed in South America, southern to the Amazonia. This charismatic group is generally diagnosed by the presence of teeth and lamellae obstructing the shell aperture (Pilsbry, 1901 [1901–1902]), but this diagnosis falls short because species of some odontostomid genera as Anctus von Martens, 1860 and some Cyclodontina Beck, 1837 have no apertural teeth. Odontostomidae is understudied, most of the genera shows a lack of clear-cut diagnostic characters, and the species composition of each genus is still a matter of controversy. The last revised family nomenclator (Bouchet et al., 2017) classified Odontostomidae with family category. However, it was hypothesized as a paraphyletic group by molecular studies (Breure, Groenenberg & Schilthuizen, 2010). Published phylogenetic hypotheses based on morphological characters are lacking for Odontostomidae.

The genus Clessinia was created by Doering in 1875 and is composed of endemic rare species from central Argentina in the Dry Chaco ecoregion. Although some other species from Brazil, such as Bulimus costatus Pfeiffer 1848, have been recently classified within the genus Clessinia (Simone, 2006; Breure & Ablett, 2012), we believe that their taxonomic assignment are not correct and should be carefully reviewed under new information. The distribution area of Clessinia overlaps largely that of Spixia Pilsbry & Vanatta, 1898, and to a lesser degree with that of Plagiodontes Doering, 1877 and Epiphragmophora Doering, 1874. The inclusion of Clessinia’s distribution area into Spixia range, their similarities in general shell shape, and a former molecular analysis (Breure & Romero, 2012) raised doubt on the monophyly of this genus. This situation was exacerbated by the fact that morphological, anatomical, and molecular studies on both genera, Clessinia and Spixia, are scarce. Moreover, most of the material kept in malacological collections generally consists in old, abraded dry shells lacking periostracum and even, when soft portions of the body are available, they are mostly not suitable for anatomical or molecular studies.

The objective of the present study was to revise the species of the genus Clessinia, under a morphological, geometric morphometrics, and molecular combined approach. One of the main questions we would like to answer is how many species composed the Clessinia cordovana complex, and how they are defined. For this, we first hypothesized that the ultrastructure of the shell periostracum will provide new taxonomic characters useful for species identification. Second, the detailed study on the genitalia of specimens from different localities over the total area of distribution will result as strong evidence for species identification. And third, that a geometric morphometric analysis, differentiating shell shape from size, will enable species separation in Clessinia. Finally, we also want to provide new molecular evidence of the species included in this work and test the validity of the genus Clessinia due to the hypothesis of paraphyly raised by Breure & Romero (2012).

Materials and Methods

Study site

The area where the study was focused is the Dry Chaco on which the majority of species of Odontostomidae are in part or completely distributed. Extending over north-central Argentina, western Paraguay, and Southeastern Bolivia, the Dry Chaco (15% of Argentina surface) is one of the largest remaining patches of forest/savanna ecosystems in Latin America (Grau et al., 2015). The Dry Chaco (225,468 km2) is dominated by deciduous forests over extensive lowland plains and mountains (below 1,600 m) with arid and semiarid climates (less than 900 mm of annual rainfall) (Izquierdo & Grau, 2009). This ecoregion is subdivided into three subecoregions: the Arid Chaco, the Semiarid Chaco, and the Chaco Serrano (Morello et al., 2012). The Semiarid Chaco extends in the middle of the Dry Chaco area and is characterized by a vegetation of semideciduous forest with shrubs and grassland that extend from the highlands to the plains. The Chaco Serrano has an open xerophytic forest with shrubs and granitic and sedimentary rocks. It extends from north to central south of Argentina in a narrow strip, subdivided into several patches in the southern part, embedded in the Arid Chaco subecoregion. The Chaco Serrano form a transition zone between humid to more xeric forests. The Arid Chaco is composed by a xeric forest with close canopy. It occupies a wide area that limits with the Monte and the Espinal ecoregions (Morello et al., 2012).

Distribution

Species distribution is based on point records (geographical coordinates) of species occurrences obtained through field work in the Córdoba and Catamarca provinces, Argentina between 2006 and 2017. All specimens collected were deposited in the Instituto de Biodiversidad Neotropical (IBN), Instituto-Fundación Miguel Lillo (IFML-MOLL), Tucumán, Argentina and the malacological collection at the Instituto de Biología Subtropical, Misiones, Argentina (IBS-Ma). Additionally, we examined other specimens and obtained records from the following collections: IBN; IFML-Moll; MACN-In, Museo Argentino de Ciencias Naturales “Bernardino Rivadavia,” Buenos Aires, Argentina (Table S1). This information was used to digitize geographical range maps and depict the extent of occurrence of each species by using QGis 2.18 (Quantum GIS Development Team, 2009; http://qgis.osgeo.org). Shapefiles layers corresponding to administrative areas of Argentina were obtained from DIVA resources (http://www.diva-gis.org/gdata) and Instituto Geográfico Nacional (http://www.ign.gob.ar/sig). Classification of Argentinean ecoregions follows Olson et al. (2001), but ecoregions and subecoregions shapefiles were obtained from ProYungas Fundation (http://siga.proyungas.org.ar/recursos).

Collecting and preservation

Hand collection of live adult specimens and dry shells of Clessinia were carried out on rocky outcrop in xerophytic areas of Córdoba and Catamarca provinces, Argentina. Specimens collected were photographed alive, then drawn in water for relaxation previous to fixation in 96% ethanol, body preservation was done using 75% ethanol. Several specimens were fixed directly in absolute ethanol, without relaxation in water, for molecular studies. Special attention was paid to the shells to preserve the periostracum structures. Shells were cleaned in an ultrasonic cleaner, air dried, and then photographed in ventral, dorsal and lateral positions, and kept in plastic boxes separated from the bodies. Photographs were taken using a Zeiss Stemi 508 with ActionCam, and measured using the software ImageJ 1.49 (Schneider, Rasband & Eliceiri, 2012; Figs. 1A–1D). Voucher specimens for the anatomical study performed were deposited at the IBN collection: IBN 886, C. cordovana (Pfeiffer, 1855); IBN 882, C. stelzneri (Doering, 1875); IBN 883, C. tulumbensis sp. nov.; IBN 890, C. pagoda Hylton Scott, 1967, and IBN 878, C. nattkemperi (Parodiz, 1944). Voucher specimens for genetic studies were: IBN 530, Spixia minor (d’Orbigny, 1837); IBN 878, C. nattkemperi; IBN 880, S. cuezzoae Salas Oroño, 2010; IBN 881, S. holmbergi (Parodiz, 1941); IBN 882, C. stelzneri; IBN 883, C. tulumbensis sp. nov.; IBN 885, Plagiodontes daedaleus (Deshayes, 1851); IBN 886, C. cordovana; IBN 890, C. pagoda.

Figure 1 Line drawings of Clessinia showing the placement of shells to obtain the linear measurements (mm).

(A) Shell ventral view. (B) Shell spire and body whorl. (C) Aperture measurements. (D) Shell lateral measurements. (E) Landmarks position in ventral view. (F) Landmarks position in lateral view. Abbreviations: Dap, apertural diameter; dm, shell minor diameter; dm1, shell minor diameter with peristome; Dm, major diameter; DS, spire width; H, total shell height; Hap, apertural height; Hbw, body whorl height; Hc, detached length; LM1-LM14, landmarks positions; S, spire height.

Morphological studies

The different zones in which the shell aperture is divided: basal, palatal (divided in upper and lower zones for internal teeth) and parietal, are the same as used by Solem (1966). Differences in terminology between a tooth and a lamella follow Cuezzo (2003). Anatomical information was obtained by dissecting 10 adult specimens per species under a Leica MZ6 stereoscope; dissected parts were illustrated with the aid of a camera lucida. Terminology for anatomical descriptions follows Tompa (1984). Terms proximal and distal refers to the position of an organ or part of an organ in relation to the gamete flow from ovotestis (proximal) to genital pore (distal), as in previous works (Cuezzo, 1997, 2006). The limit between epiphallus and penis is based on the sculpture of their inner wall. Radula, jaw and shell were observed and photographed with a SEM Zeiss Supra 55VP at the Integral Center of Electron Microscopy (CIME) of the National University of Tucumán, Argentina. The terms Diagnosis and Definition for the species description are used as established in the glossary of the International Code of Zoological Nomenclature (http://www.iczn.org).

Morphometrics

Traditional linear shell measurements were taken from specimens of each species according to availability (Fig. 1). The number of whorls was calculated following Kerney & Cameron (1979). Descriptors of measurements and proportions (mean, standard deviation, and range) were also calculated in each case. Measurements of type material of each species is recorded in the species description with the following arrange: maximum–minimum (mean) of each measurement.

Geometric morphometrics

This study was performed to quantitatively analyze the relationship between shape and size of the species of Clessinia, testing if they gave complementary information to our anatomical observations or differed from them. On these grounds, the geometric morphometric analysis was performed with 15–61 specimens per species according to their availability, totalizing 144 specimens used (Table 1). Specimens were taken from different populations ranging the whole species distribution area. Images of shell in ventral view of adult specimens were converted to TPS format with TpsUtil 1.68 (Rohlf, 2016a). Shell landmarks, discrete anatomical loci that are homologous in all individuals in the analysis, expressed by coordinates, were chosen in each case. Landmarks were located on the same shell whorl number so that comparisons among them were possible, even when shells have different whorl numbers. A total of 14 landmarks from ventral view were digitized by means of the TpsDig2 2.26 program (Rohlf, 2016b; Figs. 1E–1F). Landmarks selected in ventral view represent the general shell shape features such as body whorl, spire and aperture. A second analysis was performed using only those species of the cordovana-group from nine geographic localities to enhance the possible differences among them. Finally, another morphometric analysis was performed using six landmarks in lateral shell side (Fig. 1F) to test if the degree of detach of the aperture was significative for species delimitation. The morphometrics analyses were performed with MorphoJ 1.06d (Klingenberg, 2011). The shape symmetric components associated with position, rotation, translation, and size were removed using the Procrustes fit. A multivariate regression of the Procrustes coordinates against logarithm of centroid size, defined as square root of the sum of the squared distances of each landmark to the centroid of the landmark configuration (Bookstein, 1991), was performed to asses allometric effects (i.e., if shell shape variation is correlated with size). A permutation test was also performed with 10,000 rounds to evaluate the independence among the variables. Variation in the shell shape was examined using canonical variate analysis (CVA).

Table 1 Shell measurements among Clessinia species.

	Clessinia cordovana (n = 28)	Clessinia stelzneri (n = 25)	Clessinia tulumbensis sp. nov. (n = 61)	Clessinia pagoda (n = 15)	Clessinia nattkemperi (n = 15)	
	Mean	SD	Min	Max	Mean	SD	Min	Max	Mean	SD	Min	Max	Mean	SD	Min	Max	Mean	SD	Min	Max	
H	17.36	1.1	15.52	19.89	18.11	0.86	16.88	20.47	16.71	1.18	10.53	18.91	18.61	0.84	16.95	20.22	16.76	0.97	15.39	18.77	
Dm	3.84	0.38	3.29	4.63	4.67	0.28	4.01	5.20	4.03	0.24	3.48	4.56	5.6	0.25	5.17	5.98	4.81	0.21	4.51	5.27	
DS	4.83	0.73	3.8	6.2	3.86	0.32	3.4	4.4	4.48	0.48	3.7	5.9	4.15	0.46	3.5	5.1	4.75	0.22	4.3	5.1	
Hbw	7.97	0.49	7.24	9.34	8.59	0.4	7.89	9.39	7.52	0.34	6.67	8.34	10.75	0.52	9.98	11.75	8.35	0.43	7.74	8.93	
Dap	3.35	0.29	2.71	3.77	3.84	0.23	3.33	4.35	3.25	0.27	2.69	3.83	4.37	0.33	3.58	4.91	3.88	0.22	3.54	4.32	
Hap	4.46	0.26	3.88	4.87	4.85	0.35	4.13	5.7	4.25	0.27	3.6	4.75	6.13	0.4	5.62	6.93	5.28	0.22	4.81	5.64	
Hc	1.95	0.45	1.24	2.84	2.22	0.31	1.65	2.72	1.73	0.38	0.84	2.66	2.53	0.4	1.74	3.06	1.34	0.25	0.92	1.64	
dm	3.87	0.43	3.3	4.62	4.67	0.28	4.01	5.2	4.04	0.25	3.42	4.56	5.75	0.31	5.2	6.27	4.84	0.19	4.62	5.14	
dm1	3.96	0.48	3.31	4.76	4.39	0.45	3.46	4.86	3.77	0.31	3.02	4.38	5.78	0.42	5.04	6.65	4.3	0.35	3.75	4.94	
Notes:

Hbw, body whorl height; Hap, apertural height; H, total shell height; Dap, apertural diameter; Dm, major diameter; DS, spire width. Other variables measured on lateral view, were: shell minor diameter (dm), shell minor diameter with peristome (dm1), and detached length (Hc).

DNA extraction, polymerase chain reaction amplification, and DNA sequencing

Total DNA was extracted from three mm3 samples of foot muscle of ethanol-preserved specimens by means of a cetyltrimethylammonium bromide protocol (Beltramino et al., 2018). We selected 16 samples belonging to Clessinia and Spixia species and the outgroup species Plagiodontes daedaleus. Collection information and GenBank accession numbers for the samples analyzed are presented in Table 2. Partial sequences of the mitochondrial 16S-rRNA and the cytochrome oxidase subunit I (COI) genes, and a nuclear region including the 3′ end of the 5.8S-rRNA gene, the complete ITS-2 region, and the 5′ end of 28S-rRNA gene (hereafter referred to as ITS-2) were amplified by means of the primers 16SF-104 (5′-GAC TGT GCT AAG GTA GCA TAA T-3′) and 16SR-472 (5′-TCG TAG TCC AAC ATC GAG GTC A-3′) for 16S-rRNA (Ramírez & Ramírez, 2010), LCO1490 (5′-GGT CAA CAA ATC ATA AAG ATA TTG G-3′) and HCO2198 (5′-TAA ACT TCA GGG TGA CCA AAA AAT CA-3′) for COI (Folmer et al., 1994), and LSU-1 (5′-CTA GCT GCG AGA ATT AAT GTG A-3′) and LSU-3 (5′-ACT TTC CCT CAC GGT ACT TG-3′) for the ITS-2 (Wade & Mordan, 2000). The amplification of the 16S-rRNA gene was performed as in Rumi, Vogler & Beltramino (2017) in a T21 thermocycler (Ivema Desarrollos). The amplification of the COI gene was conducted following Vogler et al. (2014) and run on a T18 thermocycler (Ivema Desarrollos). The amplification of the ITS-2 region was performed in a total volume of 50 μl containing 30–50 ng of template DNA, each primer at 0.25 μM, 1X reaction buffer, 0.2 mM dNTPs, 2.5 mM MgCl2, and 2 U Taq Pegasus DNA polymerase (Productos Bio-Lógicos, Bernal, Argentina). The amplification was conducted in a T18 thermocycler as follows: after an initial denaturing for 3 min at 94 °C; 35 cycles of 1 min at 94 °C, 1 min at 50 °C, 1 min at 72 °C were performed; followed by a final extension at 72 °C for 5 min. The success of polymerase chain reactions (PCRs) was verified by agarose gel electrophoresis. The PCR products were purified by means of an AccuPrep PCR Purification Kit (Bioneer, Daejeon, Korea). Following purification, both DNA strands for each gene were then directly cycle-sequenced (Macrogen Inc., Seoul, South Korea). The resulting sequences were trimmed to remove the primers, and the consensus sequences between forward and reverse sequencing were obtained by means of the BioEdit 7.2.5 software (Hall, 1999). For S. minor, the repeated attempts to amplify the COI and ITS-2 regions were unsuccessful, and for this species only the 16S-rRNA was included in further analyses.

Table 2 Collection information and GenBank accession numbers for the samples used herein for the molecular studies.

Species	Geographical origin	Coordinates	Altitude	Year	Voucher	Collector	Identified by	GenBank Accession No.	
Latitude	Longitude	m.a.s.l.	COI	16S-rRNA	ITS-2	
Clessinia nattkemperi (Parodiz, 1944)	Pomancillo, at 23 km from Catamarca city (type locality), Catamarca, Argentina	−28.31219	−65.71692	652	2017	IBN 878-1	Cuezzo M.G. and Domínguez E.	Cuezzo M.G.	MG963438§	MG963450§	MH789452§	
IBN 878-2	MG963439§	MG963451§	MH789453§	
Clessinia stelzneri (Doering, 1875)	Ruta 16, Cerro San Vicente, Córdoba, Argentina	−30.42908	−64.24710	933	2017	IBN 882-1	Cuezzo M.G. and Domínguez E.	Cuezzo M.G.	MG963434§	MG963460§	MH789458§	
IBN 882-2	MG963435§	MG963461§	MH789459§	
	Dean Funes-Tulumba, Córdoba, Argentina	−30.43878	−64.28578	835	2008	IBN 560	Cuezzo M.G. and Salas Oroño E.	Cuezzo, M.G.	JF514617±	–	–	
Clessinia tulumbensis sp. nov.	Ruta 16 (between Tulumba and San José de La Dormida), Córdoba, Argentina	−30.79053	−64.63097	645	2017	IBN 883-1	Cuezzo M.G. and Domínguez E.	Cuezzo M.G.	MG963436§	MG963462§	MH789460§	
IBN 883-2	MG963437§	MG963463§	MH789461§	
	Dean Funes-Tulumba, Córdoba, Argentina	−30.40022	−64.04222	633	2008	IBN 575	Cuezzo M.G. and Salas Oroño E.	Cuezzo M.G	JF514618±	–	–	
Clessinia cordovana (Pfeiffer, 1855)	San Marcos Sierras, Córdoba, Argentina	−30.78683	−64.50069	680	2017	IBN 886-1	Cuezzo M.G. and Domínguez E.	Cuezzo M.G.	MG963446§	MG963452§	MH789462§	
IBN 886-2	MG963447§	MG963453§	MH789463§	
Clessinia pagoda Hylton Scott, 1967	San Marcos Sierras, Cerro de La Cruz, Córdoba, Argentina	−30.79722	−64.62958	832	2017	IBN 890-1	Cuezzo M.G. and Domínguez E.	Cuezzo M.G.	MG963444§	MG963456§	MH789464§	
IBN 890-2	MG963445§	MG963457§	MH789465§	
	Quilpo, Córdoba, Argentina	−30.81611	−64.64917	–	2009	RMNH 114188	Schizzi C.	–	JF514613±	–	–	
Spixia minor (d’Orbigny, 1837)	Alemanía, Quebrada de Las Conchas, Salta, Argentina	−25.62642	−65.61728	1178	2007	IBN 530-1	Salas Oroño E.	Cuezzo M.G. and Salas Oroño E.	–	MG963449§	–	
Spixia cuezzoae Salas Oroño, 2010	San Marcos Sierras, Cerro de La Cruz, Córdoba, Argentina	−30.79894	−64.62653	775	2017	IBN 880-1	Cuezzo M.G. and Domínguez E.	Cuezzo M.G.	MG963442§	MG963454§	MH789454§	
IBN 880-2	MG963443§	MG963455§	MH789455§	
Spixia holmbergi (Parodiz, 1941)	San Marcos Sierras, Córdoba, Argentina	−30.63317	−64.63317	721	2017	IBN 881-1	Cuezzo M.G. and Domínguez E.	Cuezzo M.G.	MG963440§	MG963458§	MH789456§	
IBN 881-2	MG963441§	MG963459§	MH789457§	
Plagiodontes daedaleus (Deshayes, 1851)*	Ruta 16 (between Tulumba and San José de La Dormida), Córdoba, Argentina	−30.41667	−64.07082	645	2017	IBN 885	Cuezzo M.G. and Domínguez E.	Cuezzo M.G.	MG963448§	MG963464§	MH789466§	
Clessinia gracilis Hylton Scott, 1966	Quilpo, Córdoba, Argentina	−30.81611	−64.64917	–	2009	RMNH 114228	Schizzi C.	–	JF514653±	–	–	
Spixia tucumanensis (Parodiz, 1941)	Vipos, Tucumán, Argentina	–	–	–	–	IML 15355	Cuezzo M.G	–	JF514615±	–	–	
Spixia pervarians (Haas, 1936)	Sierra de Guasapampa, Córdoba, Argentina	−30.83722	−65.34500	–	2009	RMNH 114227	Schizzi C.	–	JF514614±	–	–	
Spixia popana (Doering, 1877)	Inti Huasi-Dean Funes, Córdoba, Argentina	–	–	–	–	RMNH 114408	Schizzi C.	–	JF514616±	–	–	
Spixia philippii (Doering, 1875)	Cruz del Eje, Córdoba, Argentina	−30.75261	−64.70750	–	2009	RMNH 114226	Schizzi C.	–	JF514612±	–	–	
Notes:

IBN, Instituto de Biodiversidad Neotropical, Argentina; IML, Instituto Miguel Lillo, Argentina; RMNH, Nederlands Centrum voor Biodiversiteit (formerly Rijksmuseum van Natuurlijke Historie), The Netherlands.

* Outgroup species.

Reference to sequences: § This work; ± Breure & Romero (2012).

Sequence data, phylogenetic analyses, and molecular species delimitation

The sequence alignment of the 16S-rRNA gene was performed with MATFF 7 via the MATFF web-server (https://mafft.cbrc.jp/alignment/server/; Katoh, Rozewicki & Yamada, 2017); the COI and ITS-2 alignments were performed with Clustal X 2.1 (Larkin et al., 2007). Genetic distances among the Clessinia and Spixia species were investigated in MEGA X software (Kumar et al., 2018) using the number of differences (p) and the Kimura’s two-parameter (K2P) substitution model. Phylogenetic analyses were performed using maximum likelihood (ML), and Bayesian inference (BI). For both analyses, the COI and 16S-rRNA datasets were concatenated to improve the resolution of phylogenetic reconstructions. The total length of the analyzed matrix was 992 bp. In addition, COI-based phylogenetic reconstructions were performed to explore the phylogenetic relationships of the DNA sequences here obtained to those of other Clessinia and Spixia species from various locations available in GenBank (Table 2). The total length of this matrix was 655 bp. We also obtained phylogenetic trees for the nuclear region as an independent marker based on an 832 bp matrix. In all phylogenetic reconstructions, Plagiodontes daedaleus was used as outgroup species, with Cerion incanum (Leidy, 1851) used as an additional outgroup for the mitochondrial DNA sequence data. Sequences of C. incanum were extracted from the complete mitochondrial genome for the species (KM365085; González et al., 2016).

The ML analysis was conducted with PhyML 3.0 (Guindon et al., 2010) available via the ATGC bioinformatics platform (http://www.atgc-montpellier.fr/) with the Nearest-Neighbor Interchange branch swapping algorithm. Substitution models were selected using the SMS program (Lefort, Longueville & Gascuel, 2017) according to Akaike Information Criterion: GTR+I+G model for the concatenated dataset and the COI alignment that included GenBank sequences, and GTR+G for the ITS-2 sequences. Nodal support values were computed by bootstrapping with 1,000 replicates (Felsenstein, 1985). The BI was conducted in MrBayes 3.2.6 (Ronquist et al., 2012) with the same substitution models used in the ML analyses, as identified in jModelTest 2.1.7 (Darriba et al., 2012) by means of the corrected Akaike Information Criterion. Two runs were performed simultaneously with four Markov chains for 2 million generations, sampling every 200 generations. The first 1,001 samples of each run were discarded as burn-in, and the remaining 18,000 trees were used to estimate posterior probabilities.

The Automatic Barcode Gap Discovery (ABGD) method, which clusters sequences in putative species based on differences between intraspecific and interspecific distance variation (Puillandre et al., 2012) was used to explore species boundaries in the concatenated dataset, and the larger COI dataset including sequences from GenBank. These aligned datasets (excluding the outgroups) were analyzed via the ABGD web-server (http://wwwabi.snv.jussieu.fr/public/abgd/) using the K2P model (Vogler et al., 2016). The minimum relative gap width was set to 0.5, and the default range of prior values for maximum divergence of intraspecific diversity (p) from 0.001 to 0.1 was used. In addition, the K/θ method was used to assess the status of Clessinia species under the evolutionary genetic species concept (EGSC) (Birky et al., 2010; Birky, 2013). This method is based on basic coalescent theory and requires a phylogenetic tree as well as distance matrices to estimate the mean genetic differences within (θ) and between clades (K), in order to identify clades that are diverged enough to be considered separate species (Birky, 2013; Restrepo et al., 2014; Fontaneto, Flot & Tang, 2015). The K/θ method was performed on the concatenated dataset following Schön et al. (2012) and Birky (2013). Those clades with K/θ ratios ≥4 were considered to represent sequences that come from different evolutionary species with probability ≥0.95 (Birky, 2013 and references therein). Mean pairwise differences between clades were estimated in MEGA X.

Nomenclatural acts

The electronic version of this article in portable document format will represent a published work according to the International Commission on Zoological Nomenclature (ICZN), and hence the new names contained in the electronic version are effectively published under that Code from the electronic edition alone. This published work and the nomenclatural acts it contains have been registered in ZooBank, the online registration system for the ICZN. The ZooBank LSIDs (Life Science Identifiers) can be resolved and the associated information viewed through any standard web browser by appending the LSID to the prefix http://zoobank.org/. The LSID for this publication is: urn:lsid:zoobank.org:pub:8DB0CC34-AE26-44BA-B7F8-A5F17254BD13. The online version of this work is archived and available from the following digital repositories: PeerJ, PubMed Central, and CLOCKSS.

Results

Morphology

Periostracal ornamentation

Periostracal structures are particularly well developed in Clessinia, consisting in hairs of different lengths and densities, spines and rounded to quadrate lamellae. These structures were useful tools for species recognition. Both C. cordovana and C. stelzneri show periostracal hairs on the teleoconch surface, being notably longer among C. cordovana specimens, with more prominent hairs in specimens from Sierra de Pocho area in Córdoba. Periostracal hairs are shorter and more densely arranged in C. stelzneri. Teleoconch surface is traversed by periostracal spiral rows in the three species of the cordovana-group, with a greater number of minor spiral rows between the major hair bearing rows in C. stelzneri. In C. tulumbensis sp. nov. the periostracal hairs are absent and the spiral rows are more scatter. In Clessinia pagoda periostracal structures consist of spiral rows bearing rounded to quadrate lamellae slightly imposed over each other. In Clessinia nattkemperi lamellae are spine-shaped with wider bases almost as a triangle. All Clessinia species have an interesting pattern of periostracal microsculpture in the space between spiral rows which is traversed by axial irregular microfolds cut by spiral or diagonal microribs forming an irregular net.

Anatomy

Anatomical information obtained on the pallial, digestive and reproductive systems of each species is described in the taxonomic section.

Morphometrics

We extracted meaningful measurement differences among taxa and present these in a summary table (Table 1). In the geometric morphometric analysis performed to evaluate shell shape differences in ventral view among all Clessinia species (Fig. 2A), allometric relationships between shape and size was registered (4.94% of the total amount of shape variation; p < 0.001). The shell shape variation among the five taxa considered was successfully discriminated using CVA of the residuals from the regression of shape on centroid size. On the canonical axis 1 (CV1) (captures 74.41% of the total shell shape variation), the main changes in shell shape are associated with the expansion of the base of spire and body whorl. Specimens of C. cordovana with high scores on CV1 have thinner whorls, whereas specimens of C. nattkemperi and C. pagoda with low scores, have both spire and body whorl more expanded. It also indicates that when shells are thinner they are also taller while shells more expanded are less tall. On CV2 axis (captures 11.56% of the total shell shape variation) the main shell shape variation referred to the shape of the aperture and the degree of inclination of the suture before the aperture (landmarks 6 and 10, Fig. 1E). High scores in specimens of C. nattkemperi indicate a marked expansion in the central portion of the aperture. Specimens of C. pagoda showing low scores exhibit an oval shaped aperture, while C. tulumbensis sp. nov., C. stelzneri, and C. cordovana have intermediate forms of aperture between C. pagoda and C. nattkemperi with a higher inclination of the suture. A second analysis was performed to evaluate shell shape differences in ventral view using specimens of the cordovana species-group alone (Fig. 2B). As a result, allometric relationships between shape and size was registered (12.64% of the total amount of shape variation; p < 0.0001). Residuals from the regression of shape on centroid size were used in the analysis. On the CV1 (captures 57.49% of the total shell shape variation), the main changes in shell shape are associated with the expansion of the base of spire and body whorl. Specimens of C. cordovana from Cerro de la Cruz and the area surrounding San Marcos Sierras, plus specimens of C. tulumbensis sp. nov. from Virgen de Fatima, Route 16 and Cerro Colorado have high scores on CV1, showing thinner whorls. Specimens of C. stelzneri from Cerro San Vicente, C. tulumbensis sp. nov. from route between Dean Funes and Tulumba, Tulumba and San José de la Dormida and specimens of C. cordovana from Sierra de Pocho have low scores, showing whorls of the spire and body whorl more expanded. On CV2 axis (captures 23.66% of the total shell shape variation) the main shell shape variation is related with the shape of the aperture and the expansion of the first whorl of the shell. High scores in specimens of C. tulumbensis sp. nov. from all localities considered, except Cerro Colorado, indicate wider first whorls of the shell and a thinner aperture. Specimens of C. stelzneri and C. cordovana, with low scores, exhibit thinner fist whorl of the shell and more expanded central portion of the aperture. C. tulumbensis sp. nov. from Cerro Colorado have intermediate forms between both previous described groups. Analysis using landmarks in shell lateral views did not show significative differences among the species (Fig. S1).

Figure 2 Geometric morphometric analyses.

Canonical variate analyses (CVA) of shell shape variation (ventral view) along the first two canonical axes. Wireframe diagrams show shape changes associated with variation along each axis. (A) Based on all Clessinia species. (B) based on Clessinia cordovana species group from different localities of occurrences.

Molecular analyses

Sequence data, phylogenetic analyses, and molecular species delimitation

We successfully amplified both mitochondrial loci and the nuclear region in the majority of Clessinia and Spixia specimens, except for S. minor in which amplification of the COI and ITS-2 markers was not possible. Partial 16S-rRNA sequences ranged between 287 and 295 bp, COI sequences consisted of 655 bp, and ITS-2 sequences were of 822 bp in length for all individuals. The ITS-2 region showed no sequence variation within each species and exhibited little genetic differentiation among species (Tables 3 and 4). Phylogenetic reconstructions obtained with the nuclear marker were unresolved (Fig. S2). For the mitochondrial markers, ML and BI results revealed congruent topologies; consequently, we reported only the BI tree. From the analyses of the concatenated dataset, Clessinia stelzneri clustered with C. tulumbensis sp. nov.; similarly, S. cuezzoae clustered with specimens of Clessinia pagoda, and this group clustered with S. holmbergi. C. cordovana clustered with the group formed by these three species (Fig. 3). Therefore, these trees did not support the monophyly of Clessinia (Fig. 3). The phylogenetic trees inferred from the larger COI dataset including sequences from GenBank congruently identify roughly the same major groups, with both genera being paraphyletic due to association of Clessinia and Spixia specimens in well-supported arrangements, as shown by the relationships between C. nattkemperi and S. tucumanensis (Parodiz, 1941) or S. cuezzoae and C. pagoda (Fig. 4). Sequence divergence for the mitochondrial loci amongst the species are presented in Tables 5 and 6.

Table 3 Polymorphic positions based on a 822 bp DNA fragment of the 5.8S-ITS2-28S region for Clessinia and Spixia species.

	113	117	140	233	310	311	361	469	516	712	
C. tulumbensis sp. nov.	C	C	A	G	A	G	C	A	A	C	
C. stelzneri	·	·	·	·	·	·	·	·	·	·	
C. cordovana	A	A	·	·	G	·	·	·	·	·	
C. nattkemperi	A	·	·	·	·	T	T	G	·	·	
C. pagoda	A	·	·	·	·	·	·	·	·	·	
S. cuezzoae	A	·	C	·	·	·	·	·	·	·	
S. holmbergi	A	·	·	T	·	·	·	·	T	T	
Note:

Numbers indicate the position of variable sites. C. tulumbensis sp. nov. is shown as reference sequence; dot indicates identity with the reference sequence.

Table 4 Genetic distances in ITS-2 sequences among Clessinia and Spixia species.

Species	GenBank No.*	ID	1	2	3	4	5	6	7	8	9	10	11	12	13	14	
C. nattkemperi	MH789452	1	–	0.000	0.006	0.006	0.005	0.005	0.004	0.004	0.007	0.007	0.005	0.005	0.005	0.005	
	MH789453	2	0.000	–	0.006	0.006	0.005	0.005	0.004	0.004	0.007	0.007	0.005	0.005	0.005	0.005	
C. cordovana	MH789462	3	0.006	0.006	–	0.000	0.004	0.004	0.002	0.002	0.006	0.006	0.004	0.004	0.004	0.004	
	MH789463	4	0.006	0.006	0.000	–	0.004	0.004	0.002	0.002	0.006	0.006	0.004	0.004	0.004	0.004	
S. cuezzoae	MH789454	5	0.005	0.005	0.004	0.004	–	0.000	0.001	0.001	0.005	0.005	0.002	0.002	0.002	0.002	
	MH789455	6	0.005	0.005	0.004	0.004	0.000	–	0.001	0.001	0.005	0.005	0.002	0.002	0.002	0.002	
C. pagoda	MH789464	7	0.004	0.004	0.002	0.002	0.001	0.001	–	0.000	0.004	0.004	0.001	0.001	0.001	0.001	
	MH789465	8	0.004	0.004	0.002	0.002	0.001	0.001	0.000	–	0.004	0.004	0.001	0.001	0.001	0.001	
S. holmbergi	MH789456	9	0.007	0.007	0.006	0.006	0.005	0.005	0.004	0.004	–	0.000	0.005	0.005	0.005	0.005	
	MH789467	10	0.007	0.007	0.006	0.006	0.005	0.005	0.004	0.004	0.000	–	0.005	0.005	0.005	0.005	
C. stelzneri	MH789458	11	0.005	0.005	0.004	0.004	0.002	0.002	0.001	0.001	0.005	0.005	–	0.000	0.000	0.000	
	MH789459	12	0.005	0.006	0.004	0.004	0.002	0.002	0.001	0.001	0.005	0.005	0.000	–	0.000	0.000	
C. tulumbensis sp. nov.	MH789460	13	0.005	0.005	0.004	0.004	0.002	0.002	0.001	0.001	0.005	0.005	0.000	0.000	–	0.000	
	MH789461	14	0.005	0.005	0.004	0.004	0.002	0.002	0.001	0.001	0.005	0.005	0.000	0.000	0.000	–	
Notes:

Uncorrected (below the diagonal) and corrected (K2P; above the diagonal) distances are shown.

* References to the sequences are provided in Table 2.

Table 5 Genetic distances in 16S-rRNA sequences among Clessinia and Spixia species.

Species	GenBank No.*	ID	1	2	3	4	5	6	7	8	9	10	11	12	13	14	15	
S. minor	MG963449	1	–	0.163	0.163	0.167	0.159	0.176	0.176	0.162	0.162	0.166	0.171	0.221	0.221	0.226	0.221	
C. nattkemperi	MG963450	2	0.145	–	0.018	0.162	0.166	0.157	0.157	0.153	0.153	0.175	0.179	0.198	0.198	0.198	0.193	
	MG963451	3	0.145	0.018	–	0.149	0.153	0.157	0.157	0.153	0.153	0.184	0.188	0.198	0.198	0.198	0.193	
C. cordovana	MG963452	4	0.149	0.145	0.135	–	0.007	0.158	0.158	0.149	0.149	0.180	0.184	0.193	0.193	0.198	0.193	
	MG963453	5	0.142	0.149	0.138	0.007	–	0.158	0.158	0.149	0.149	0.180	0.184	0.193	0.193	0.198	0.193	
S. cuezzoae	MG963454	6	0.156	0.142	0.142	0.142	0.142	–	0.000	0.036	0.036	0.131	0.136	0.202	0.207	0.207	0.202	
	MG963455	7	0.156	0.142	0.142	0.142	0.142	0.000	–	0.036	0.036	0.131	0.136	0.202	0.207	0.207	0.202	
C. pagoda	MG963456	8	0.145	0.138	0.138	0.135	0.135	0.035	0.035	–	0.000	0.136	0.140	0.189	0.193	0.193	0.189	
	MG963457	9	0.145	0.138	0.138	0.135	0.135	0.035	0.035	0.000	–	0.136	0.140	0.189	0.193	0.193	0.189	
S. holmbergi	MG963458	10	0.149	0.156	0.163	0.160	0.160	0.121	0.121	0.124	0.124	–	0.007	0.193	0.198	0.198	0.193	
	MG963459	11	0.152	0.160	0.167	0.163	0.163	0.124	0.124	0.128	0.128	0.007	–	0.193	0.198	0.198	0.193	
C. stelzneri	MG963460	12	0.191	0.174	0.174	0.170	0.170	0.177	0.177	0.167	0.167	0.170	0.170	–	0.011	0.014	0.011	
	MG963461	13	0.191	0.174	0.174	0.170	0.170	0.181	0.181	0.170	0.170	0.174	0.174	0.011	–	0.018	0.014	
C. tulumbensis sp. nov.	MG963462	14	0.195	0.174	0.174	0.174	0.174	0.181	0.181	0.170	0.170	0.174	0.174	0.014	0.018	–	0.004	
	MG963463	15	0.191	0.170	0.170	0.170	0.170	0.177	0.177	0.167	0.167	0.170	0.170	0.011	0.014	0.004	–	
Notes:

Uncorrected (below the diagonal) and corrected (K2P; above the diagonal) distances are shown.

* References to the sequences are provided in Table 2.

Table 6 Genetic distances in COI sequences among Clessinia and Spixia species.

Species	GenBank No.*	ID	1	2	3	4	5	6	7	8	9	10	11	12	13	14	15	16	17	18	19	20	21	22	
C. nattkemperi	MG963438	1	–	0.019	0.132	0.163	0.163	0.164	0.169	0.156	0.164	0.173	0.170	0.165	0.152	0.147	0.157	0.149	0.151	0.156	0.158	0.166	0.164	0.156	
	MG963439	2	0.019	–	0.136	0.170	0.170	0.176	0.181	0.159	0.171	0.178	0.175	0.175	0.156	0.152	0.167	0.157	0.159	0.164	0.166	0.173	0.166	0.159	
S. tucumanensis	JF514615	3	0.119	0.123	–	0.165	0.168	0.167	0.164	0.161	0.171	0.168	0.175	0.173	0.175	0.166	0.174	0.190	0.187	0.192	0.190	0.195	0.192	0.197	
S. cuezzoae	MG963442	4	0.145	0.151	0.147	–	0.002	0.052	0.050	0.052	0.144	0.161	0.161	0.151	0.156	0.153	0.170	0.147	0.147	0.150	0.151	0.149	0.152	0.147	
	MG963443	5	0.145	0.151	0.149	0.002	–	0.054	0.052	0.054	0.147	0.163	0.163	0.153	0.153	0.151	0.168	0.147	0.147	0.150	0.151	0.149	0.152	0.147	
C. pagoda	MG963444	6	0.145	0.155	0.147	0.050	0.052	–	0.009	0.042	0.136	0.164	0.166	0.149	0.157	0.150	0.176	0.155	0.157	0.157	0.159	0.154	0.160	0.152	
	MG963445	7	0.149	0.158	0.145	0.048	0.050	0.009	–	0.044	0.136	0.164	0.166	0.149	0.162	0.154	0.176	0.157	0.160	0.160	0.162	0.157	0.157	0.155	
	JF514613	8	0.140	0.142	0.143	0.050	0.052	0.041	0.043	–	0.136	0.157	0.154	0.142	0.140	0.138	0.152	0.145	0.145	0.148	0.150	0.152	0.148	0.141	
S. philippii	JF514612	9	0.145	0.151	0.151	0.130	0.132	0.123	0.123	0.123	–	0.183	0.173	0.159	0.164	0.159	0.184	0.183	0.190	0.195	0.188	0.192	0.193	0.187	
S. holmbergi	MG963440	10	0.153	0.156	0.149	0.143	0.145	0.145	0.145	0.140	0.160	–	0.021	0.118	0.158	0.148	0.168	0.175	0.173	0.173	0.172	0.175	0.178	0.175	
	MG963441	11	0.151	0.155	0.155	0.143	0.145	0.147	0.147	0.138	0.153	0.020	–	0.125	0.153	0.144	0.160	0.172	0.170	0.170	0.170	0.167	0.175	0.172	
S. pervarians	JF514614	12	0.147	0.155	0.153	0.136	0.138	0.134	0.134	0.128	0.142	0.108	0.114	–	0.139	0.130	0.158	0.170	0.175	0.175	0.172	0.184	0.160	0.158	
C. cordovana	MG963446	13	0.136	0.140	0.155	0.140	0.138	0.140	0.143	0.127	0.145	0.140	0.136	0.125	–	0.015	0.089	0.168	0.170	0.175	0.175	0.190	0.168	0.165	
	MG963447	14	0.132	0.136	0.147	0.138	0.136	0.134	0.138	0.125	0.142	0.132	0.128	0.117	0.015	–	0.076	0.161	0.163	0.168	0.168	0.182	0.165	0.163	
C. gracilis	JF514653	15	0.140	0.147	0.153	0.151	0.149	0.155	0.155	0.136	0.160	0.147	0.142	0.140	0.082	0.071	–	0.165	0.165	0.168	0.167	0.167	0.172	0.175	
C. stelzneri	MG963434	16	0.134	0.140	0.166	0.132	0.132	0.138	0.140	0.130	0.160	0.155	0.153	0.151	0.149	0.143	0.147	–	0.006	0.013	0.015	0.023	0.025	0.027	
	MG963435	17	0.136	0.142	0.164	0.132	0.132	0.140	0.142	0.130	0.166	0.153	0.151	0.155	0.151	0.145	0.147	0.006	–	0.013	0.015	0.023	0.027	0.029	
C. tulumbensis sp. nov.	MG963436	18	0.140	0.145	0.168	0.134	0.134	0.140	0.142	0.132	0.169	0.153	0.151	0.155	0.155	0.149	0.149	0.013	0.013	–	0.009	0.017	0.031	0.033	
	MG963437	19	0.142	0.147	0.166	0.136	0.136	0.142	0.143	0.134	0.164	0.153	0.151	0.153	0.155	0.149	0.149	0.015	0.015	0.009	–	0.019	0.033	0.032	
	JF514618	20	0.147	0.153	0.169	0.134	0.134	0.138	0.140	0.136	0.168	0.155	0.149	0.162	0.166	0.160	0.149	0.022	0.022	0.017	0.019	–	0.041	0.042	
C. stelzneri	JF514617	21	0.145	0.147	0.168	0.136	0.136	0.142	0.140	0.132	0.168	0.156	0.155	0.143	0.149	0.147	0.153	0.024	0.026	0.030	0.032	0.039	–	0.009	
S. popana	JF514616	22	0.140	0.142	0.171	0.132	0.132	0.136	0.138	0.127	0.164	0.155	0.153	0.142	0.147	0.145	0.155	0.026	0.028	0.032	0.032	0.041	0.009	–	
Notes:

Uncorrected (below the diagonal) and corrected (K2P; above the diagonal) distances are shown.

* References to the sequences are provided in Table 2.

Figure 3 Bayesian tree of Clessinia and Spixia species based on a 992 bp multilocus dataset (COI and 16S-rRNA).

The posterior probability values for BI and bootstrap values for the ML tree are shown above and below the branches. Numbers within groups are GenBank accession numbers. Gray bars indicate putative species identified by the ABGD analysis.

Figure 4 Bayesian tree of Clessinia and Spixia species based on the partial COI gene.

The posterior-probability values for BI and bootstrap values for the ML tree are shown above and below the branches. Numbers within groups are GenBank accession numbers. Gray bars indicate putative species identified by the ABGD analysis.

By using the concatenated dataset, the ABGD approach recovered six candidate species based on the distribution of the pairwise genetic distances with a maximum prior of intraspecific divergence of 0.035938 (Fig. 3). The same as the ABGD, the K/θ method provided support for six of the morphospecies to be considered different evolutionary genetic species (Table 7), except for C. stelzneri and C. tulumbensis sp. nov. which were not supported as distinct genetic species by either method. Based on the COI dataset, the ABGD analysis clustered sequences into 10 stable putative species based on the distribution of the pairwise genetic distances with a maximum prior of intraspecific divergence of 0.035938. The species C. stelzneri, C. tulumbensis sp. nov., and S. popana (Doering, 1875 [1877a]) were clustered within the same group. All the remaining species, that is, C. nattkemperi, S. tucumanensis, S. cuezzoae, C. pagoda, S. philippii (Doering, 1875), S. holmbergi, S. pervarians (Haas, 1936), C. cordovana, and C. gracilis Hylton Scott, 1966 were assigned to different candidate species (Fig. 4).

Table 7 Summary of results of K/θ tests for Clessinia and Spixia species based on a multi-locus dataset (COI and 16S-rRNA).

Groups tested	θ1 and θ2	K	K/θ ratio	n1, n2	
C. stelzneri—C. tulumbensis sp. nov.	0.01945946
0.01286174	0.017108882	0.88
1.33	2, 2	
C. pagoda—S. cuezzoae	0.01082642
0.00214056	0.0464957308	4.29
21.72	2, 2	
C. pagoda—S. holmbergi	0.01082642
0.03578084	0.1481874665	13.69
4.14	2, 2	
S. holmbergi—S. cuezzoae	0.03578084
0.00214056	0.1502558675	4.20
70.19	2, 2	
C. cordovana—S. cuezzoae	0.03344482
0.00214056	0.1532695330	4.58
71.60	2, 2	
C. cordovana—S. holmbergi	0.03344482
0.03578084	0.1648849724	4.93
4.61	2, 2	
C. cordovana—C. pagoda	0.03344482
0.01082642	0.1502652394	4.49
13.88	2, 2	
C. nattkemperi—C. cordovana	0.04049494
0.03344482	0.1573729569	3.89
4.70	2, 2	
C. nattkemperi—S. cuezzoae	0.04049494
0.00214056	0.1649819496	4.07
77.07	2, 2	
C. nattkemperi—S. holmbergi	0.04049494
0.03578084	0.1733817044	4.28
4.85	2, 2	
C. nattkemperi—C. pagoda	0.04049494
0.01082642	0.1616734496	3.99
14.93	2, 2	
C. stelzneri and C. tulumbensis sp. nov.—C. nattkemperi	0.01921596
0.04049494	0.1739526225	9.05
4.30	4, 2	
C. stelzneri and C. tulumbensis sp. nov.—C. cordovana	0.01921596
0.03344482	0.1776448056	9.24
5.31	4, 2	
C. stelzneri and C. tulumbensis sp. nov.—S. cuezzoae	0.01921596
0.00214056	0.1687162889	8.78
78.82	4, 2	
C. stelzneri and C. tulumbensis sp. nov.—S. holmbergi	0.01921596
0.03578084	0.1810391204	9.42
5.06	4, 2	
C. stelzneri and C. tulumbensis sp. nov.—C. pagoda	0.01921596
0.01082642	0.1706482546	8.88
15.76	4, 2	
Note:

θ, mean pairwise sequence difference within a clade; K, mean pairwise sequence difference between clades; n1, n2, number of sequences within each of the clades compared.

As a result of the anatomical studies performed, shell periostracum observations, shell geometric morphometrics and genetic analyses, and based on previous findings (Breure & Romero, 2012), we here synonymized the genus Clessinia and Spixia and according to the principle of priority (ICZN Code, Art.23.1) the valid name of the taxon should be Clessinia Doering, 1875 which has priority over Spixia Pilsbry & Vanatta, 1894. In the following, we provide the taxonomic description and new systematic arrangement of the treated species.

Taxonomic descriptions

Superfamily Orthalicoidea Martens, 1860

Family Odontostomidae Pilsbry & Vanatta, 1898

Genus Clessinia (Doering, 1875)

Bulimus “Clessinia” Doering, (1874 [1875]): 201.

Bulimus “Macrodontes”–Doering, 1875 [1877a]: 331;–Doering, 1875 [1877b]: 250.

Scalarinella Doering–in Dohrn, 1875: 202.

Odontostomus (Scalarinella)–Pilsbry, 1901 [1901–1902]: 66;–Parodiz, 1939: 731.

Cyclodontina (Clessinia)–Parodiz, 1944;–Hylton Scott, 1966: 30.

Clessinia Doering–Hylton Scott, 1967: 103;–Fernández, 1973: 142;–Breure, 1974: 110;–Cuezzo, Miranda & Ovando, 2013.

Odontostomus (Spixia) Pilsbry & Vanatta, 1898: 57 [new synonymy]

Type species. Clessinia stelzneri (Doering, 1875).

Definition. Shell fusiform to turritelliform. Protoconch with delicate axial ribs and spiral bands delimited by thin grooves. Shell with periostracal complex structures consisting on spiral rows bearing “hairs” or triangular, rectangular to quadrate lamellae. Few species lacking periostracal ornamentation. Last portion of body whorl with aperture with slightly reflexed peristome, some species forming a cornet detached from rest of shell body whorl with peristome thin and expanded. Body whorl microsculpture complex, consisting in microfolds forming an irregular net, which in some cases is expanded dorsally over shell cornet. Fourth to five inner apertural teeth forming a complex apertural barrier, except for one species with three teeth. Dorsal portion of shell body whorl with a medial marked notch corresponding to the basal lamella. Columellar lamella undulating, in some species L-shaped. Presence of a short penial sheath overlapping distal portion of the penis. Insertion of penial muscle at proximal penis or distal epiphallus. Vas deferens thin, running freely along penis and attached to penial retractor muscle.

Diagnosis. Clessinia is one of the odontostomid groups showing most complex apertural teeth arrangements. Number of apertural teeth/lamellae ranges from three to five. Together with Plagiodontes (type species = Helix dentata Wood, 1828) has a similar protoconch sculpture consisting on axial ribs and transversal grooves between ribs. It differs from Plagiodontes in general shell shape, showing thinner and taller spires and larger numbers of whorls plus strong differences in number of apertural teeth/lamellae. Species here redescribed plus the new species show a shell aperture detached from the body whorl and this character is not observed in the remaining species of Clessinia (former Spixia) and in no other odontostomid genus. Shell aperture shape varies from subcircular to subquadrate.

Clessinia is found in dry habitats with its distribution area ranging from Argentina, Uruguay to Bolivia and Paraguay. Clessinia differs from Cyclodontina (type species = Clausilia pupoides Spix, 1827) (Cowie, Cazzaniga & Glaubrecht, 2004) in having more apertural teeth/lamellae, some species of Cyclodontina are even toothless. Shells of Cyclodontina are basally wider, sometimes glossy, without any particular ultrastructural shell sculpture described for this genus. The shell aperture in Cyclodontina is not detached from the body whorl as in some Clessinia species. Cyclodontina is distributed in Bolivia, Paraguay, Argentina, Uruguay, and Brazil. Clessinia differs from Pilsbrylia (type species = Pilsbrylia paradoxa Hylton Scott, 1952), in the general shell shape because Pilsbrylia species have a fusiform, broader shell shape, and the presence of only two apertural teeth. On the contrary to Clessinia, Pilsbrylia species inhabit in humid forest.

Habitat preferences. Species of Clessinia inhabit dry areas where rocky formations are frequently found among low xerophytic vegetation. Few species occur in Yungas ecoregion, but in transition zones with dryer forests. They usually live below rocks in contact to the ground, in rock crevices, or buried in soil under shrubs. Some species can be found glued to leaves in bushes. Clessinia nattkemperi is usually found attached to the surface of cactuses or under dead cactus in contact with soil.

Species distribution (Figs. 5A–5D)

Figure 5 Distribution of Clessinia species.

(A) Position of Argentina in South America. (B) Ecoregions in north-central Argentina, note that the Dry Chaco ecoregion area is highlighted with a brown line limits. Quadrate areas correspond to (C and D) figures; (C) Catamarca province with localities of occurrences for C. nattkemperi and C. cordovana, ecoregions colors as in (B); (D) Northern Córdoba province with localities of occurrences of resting Clessinia species, ecoregions colors as in (B).

Clessinia species here treated are distributed in the Pampean Sierras of Central Argentina, in the portion corresponding to the provinces of Córdoba and Catamarca (Fig. 5A). These Sierras form a mountain complex of about 300 km2 in extent with a direction of north to south and consist in a series of parallel mountain ranges. An extended depression of salty surface called Salinas Grandes, located between northern Córdoba, southeastern Catamarca and La Rioja provinces, and Salinas de Ambargasta between southern Santiago del Estero and northwestern Córdoba, subdivide the Pampean Sierras forming a real ecological barrier for land snail dispersion (Figs. 5B and 5D). Main mountain systems in Córdoba are the Sierra Chica, Sierra Grande and Sierra de Comechingones, this last is extended to San Luis province. Clessinia is mainly distributed around and to the north of the Sierra Chica, including minor mountains such as Sierras de Ischilin, Higuerita, Copacabana, and Massa in Córdoba. Also scatter occurrences have been registered to the southwest in Sierra de Pocho. In Catamarca, occurrences are registered in the Ambato and Esquiu departments, both also corresponding to the Pampean Sierras but to the northwest of the Salinas Grandes (Figs. 5B and 5C). All the localities of occurrences of Clessinia species here considered are found in different patches areas of Chaco Serrano between 400 and 1,500 m above sea level. Remaining Clessinia species (former genus Spixia) have a wider area of distribution in the Dry Chaco, Espinal and Monte ecoregions in Argentina (Salas Oroño, 2007, 2010) (Fig. 5B).

Species description

Clessinia cordovana (Pfeiffer, 1855)

Figs. 5–9, Tables 1, 2.

Bulimus cordovanus Pfeiffer, 1855: 149;–Pfeiffer, 1856: 34;–Pfeiffer, 1859: 435;–Dohrn, 1875: 202; 1877: 157;–Kobelt, 1878: 150;–Von Martens, 1890–1891: 251;–Breure, 1974: 114.

Bulimus “Macrodontes” cordovanus–Doering, 1875 [1877a]: 331;–Doering, 1875 [1877b]: 250.

Odontostomus (Scalarinella) cordovanus–Pilsbry, 1901 [1901–1902]: 66, pl. 13, fig. 100.

Odontostomus (Macrodontes) cordovanus–Holmberg, 1912: 152.

Odontostomus (Scalarinella) cordovanus–Parodiz, 1939: 732, fig. 1.

Cyclodontina (Scalarinella) cordovana–Parodiz, 1944: 5;–1957: 29.

Cyclodontina (Clessinia) cordovanus–Hylton Scott, 1966: 31, figs. 1–5, 7.

Cyclodontina (Clessinia) gracilis Hylton Scott, 1966: 34, figs. 6, 8.

Clessinia cordovana–Fernández, 1973: 142;–Cuezzo, Miranda & Ovando, 2013: 28.

Clessinia gracilis–Fernández, 1973: 144; Breure & Romero, 2012: 18.

Cyclodontina (Clessinia) gracilis–Breure, 1974: 116.

Type material. Lectotype SMF 10417a (H: 16.3; Dap: 3.7; Dm 4.5; Hap: 4.8); paralectotype SMF 10417b (H: 17; Dap: 3.6; Dm: 4.7; Hap: 4.4); holotype Cyclodontina (Clessinia) gracilis Hylton Scott, 1966, MACN-In 6421.

Type locality. Argentina, Córdoba province. “Pendiente Oeste de la Sierra de Aconjigasta, en las quebradas húmedas como la de la Mermela, de Jatan, del Nieve y más al sud cerca de Aguas de los Oscuros” (Doering, 1875 [1877a]).

Description

External features (Figs. 6A–6C): Body dark to light grey with two blackish pigmented, longitudinal bands extending from the mantle collar to the tentacles. Black tentacles. Foot short, light gray, with a blunt end.

Figure 6 Clessinia cordovana, general shell morphology and habitat.

(A and B) Live specimen from Sierra de Pocho, central Córdoba. (C) Live specimen from San Marcos Sierras, northwestern Córdoba. (D) Ventral, (E) lateral, and (F) dorsal views of a shell from Sierra de Pocho, note the length of the periostral hairs, scale bar = four mm (IFML-Moll 15415). (G) Ventral, (H) lateral, and (I) dorsal views of a shell from San Marcos Sierras, note that the length of the periostracal hairs is shorter than in the previous locality, scale bar = five mm (IBN 563. (J) Oval-shaped shell aperture of C. cordovana. (K) Square-shaped shell aperture. (L) Semicircular shaped shell aperture. (M) Part of the body whorl and aperture showing inner portion of lower columellar lamella. (N) Detail of dorsal view of the body whorl. (O) View of the species habitat in San Marcos Sierras, Córdoba. (P) View of the species habitat in Sierra de Pocho, Córdoba. Photographs by M.G. Cuezzo.

Shell (Figs. 6A–6L and 7): Turritelliform to subfusiform, comprising 8 ½ to 9 ½ slightly convex whorls. Coloration pale to dark brown, uniform (Figs. 6A–6I). Protoconch with axial, regularly arranged strength ribs, and thin spiral parallel bands delimited by spiral grooves between ribs (Figs. 7A and 7B). Teleoconch with axial, oblique, shallow thin costules separated by regular spaces. Surface of the teleoconch traversed by spiral rows bearing two types of periostracal hairs (Figs. 7C–7F). Spiral rows bearing long hairs of 200–300 μm (mean = 227, n = 8) intercalated with two to three spiral rows, one of each bearing hairs wider at base and less tall (Fig. 7E). Departing from each spiral row, interconnected axial microfolds giving the appearance of an irregular net. Suture deeply impressed. Distal portion of body whorl detached from rest of the shell forming a cornet (Figs. 6D, 6E, 6G, 6H and 7C). Aperture suboval, round to square with thin, continuous, expanded peristome. Five inner lamellae in the aperture not connecting to the peristome (Figs. 6J–6L). Upper columellar lamella long, straight, spirally following the columellar axis. Lower columellar lamella running parallel, slightly undulating, spirally following columellar axis (Fig. 6M). Basal teeth straight, short, to the left of the aperture producing a groove on dorsal side of the shell (Fig. 6F). Some specimens with dorsal groove not marked (Fig. 6N). Upper palatal teeth small, generally triangular shaped. Lower palatal teeth short. Both palatal teeth perpendicular to columellar axis, deeply located inside cornet. Dorsal side of the aperture with an inner marked groove (Figs. 6J–6L). Umbilicus narrow. Shell measurements represented in Table 1.

Figure 7 Clessinia cordovana, shell ultrastructure.

(A) Protoconch sculpture with axial, regularly arranged strength ribs, scale bar = 100 μm. (B) Detail of protoconch showing thin spiral parallel bands delimited by spiral grooves between ribs, scale bar = 10 μm. (C) Body whorl and aperture detached from the rest of the shell, showing the long periostracal hairs, scale bar = 1,000 μm. (D) Conic spire with spiral lines bearing hears, scale bar = 1,000 μm. (E) Detail of periostracal hairs with triangular base, scale bar = 100 μm. (F) Detail of the interconnected axial microfolds giving the appearance of an irregular net, scale bar = 100 μm. Photographs by M.G. Cuezzo.

Jaw (Fig. 8A): Wide horseshoe shaped. Ten plaques with a triangular central one subdivided into three subplaques, the middle one more triangular-shaped. Five lateral quadrangular to rectangular shaped plaques at both sides of the central one. Lateral plaques slightly increasing their size toward the tip of the horseshoe. Each plaque traversed by several thin transversal grooves.

Figure 8 Radula and jaw in Clessinia.

(A) Jaw of C. cordovana, scale bar = 100 μm. (B) Jaw of C. stelzneri, scale bar = 100 μm. (C) Jaw of C. tulumbensis sp. nov., scale bar = 100 μm. (D) Jaw of C. pagoda, scale bar = 100 μm. (E) Jaw of C. nattkemperi, scale bar =100 μm. (F–I) C. stelzneri: (F) General view of the radula, scale bar = 10 μm. (G) Central and first lateral teeth, scale bar = 10 μm. (H) Lateral and marginal teeth in general view, scale bar = 10 μm. (I) Detail of lateral teeth, scale bar = 10 μm.

Pallial system: Pulmonary roof thin and long traversed by few veins mostly concentrated on distal portion. Kidney triangular, short, of a quarter of the total length of the pulmonary roof. Secondary ureter closed over most of its length, opening slightly before rectum. Pallial gland thin, parallel to mantle collar. Afferent vein parallel to main pulmonary vein.

Reproductive system (Figs. 9A–9E): Ovotestis embedded into digestive gland within the fourth and fifth shell whorls. Hermaphroditic duct inserting in distal portion of albumen gland (Fig. 9A). Seminal receptacle swollen. Fertilization pouch-spermathecal complex long, digitiform broaden at its base. Bursa copulatrix with sac rounded, longer than spermoviduct reaching the albumen gland. External limits between epiphallus and penis not evident (Figs. 9A and 9B). Penis cylindrical, long, with a short penis sheath overlapping in part distal penis. Inner morphology of the penis divided into three areas marked by differential pattern of sculpture (Figs. 9C and 9D). Proximal portion with same diameter than resting portions, inner sculpture with tightly appressed pustules (Fig. 9E). Penial papilla absent. Penis medial sector long, cylindrical, inner wall with rhomboidal to hexagonal pustules covering the surface, without pilasters (Figs. 9D and 9E). Distal penis short, inner sculpture consisting in three to four longitudinal, straight, thin pilasters, parallel to each other (Figs. 9C and 9D). Penial retractor muscle short and thick, inserting in penis proximal portion. Epiphallus ¼ of penial length. Flagellum thinner than epiphallus and ½ epiphallus length. Vas deferens thin, running freely along penis, attached to penial retractor muscle, then free along epiphallus and inserting between flagellum and epiphallus. Vagina cylindrical, with a distal portion thinner in diameter than the proximal, inner wall with longitudinal pilasters. Vagina about double in length of the distal portion of penis.

Figure 9 Clessinia cordovana, anatomy.

(A) General view of the reproductive system dissected out, limits penis/epiphallus is indicated, scale bar = two mm. (B) Detail of the proximal portion of the penis complex showing the vas deferens attached to the penis muscular retractor, scale bar = two mm. (C) Inner sculpture of penis showing the limits of three areas of the penial complex, note the position of the vas deferens, scale bar = two mm. (D) Photograph of the inner sculpture of penis, same scale bar equal to figure C. (E) Detail of the rhomboidal pustules located in the medial inner portion of the penis wall. Abbreviations: ag, albumen gland; bc, bursa copulatrix; dp, penis distal portion; e, epiphallus; f, flagellum; hd, hermaphroditic duct; mp, penis medial portion; mr, penial retractor muscle; p, penis; ppr, penis proximal portion; ps, penis sheath; v, vagina; vd, vas deferens.

Habitat (Figs. 6O and 6P): Calcareous rocky outcrops on mountain slope, under and between roots of woody shrubs.

Distribution (Figs. 5C and 5D): Disjunct distribution between Córdoba and Catamarca provinces. Northwestern mountain ranges of Córdoba province, in Cruz del Eje, Punilla, Pocho, and Tulumba departments. Clessinia gracilis was described in 1966 from a single shell found in La Puerta, Ambato department, Catamarca province, and was synonymized to C. cordovana (Cuezzo, Miranda & Ovando, 2013) because the holotype has same size and shape as C. cordovana. However, during different collecting trips to the area of La Puerta carried out during summer in different years, specimens were not found. This species inhabits the Dry Chaco ecoregion, Chaco Serrano subecoregion.

Clessinia stelzneri (Doering, 1875)

Figures 2, 5 and 10–12; Tables 1 and 2.

Bulimus (Clessinia) stelzneri Doering, 1874 [1875]: 201.

Bulimus “Macrodontes” cordovanus var. stelzneri–Doering, 1875 [1877a]: 332;–Doering, 1875 [1877b]: 251.

Odontostomus (Scalarinella) cordovanus var. stelzneri–Pilsbry, 1901 [1901–1902]: 67.

Odontostomus (Scalarinella) cordovanus stelzneri–Parodiz, 1939: 732, fig. 2.

Scalarinella (Scalarinella) cordovana stelzneri–Zilch, 1959–1960: 508.

Cyclodontina (Scalarinella) cordovanus stelzneri–Parodiz, 1957: 29.

Scalarinella (Scalarinella) cordovana stelzneri–Zilch, 1971: 198, pl. 12, fig. 14;–Neubert & Janssen, 2004: 230, pl. 19, fig. 248.

Clessinia cordovana stelzneri–Breure, 1974: 110;–Breure & Schouten, 1985: 9, fig. 3.

Clessinia stelzneri–Cuezzo, Miranda & Ovando, 2013: 29.

Type material. Lectotype SMF 10417/3a; paralectotypes SMF 26582 (1), SMF 26583 (2), SMF 325584 (4).

Type locality. “… quebrada de Yatan (Serrezuela; Provincia de Córdova).” According to Hylton Scott (1966) the type locality is located in Argentina, Córdoba Prov., Cruz del Eje Dept., Yatan, Serrezuela.

Description

External features (Fig. 10A): Body light brown. Foot short with a blunt end. Some specimens with sole lighter than dorsal body coloration.

Figure 10 Clessinia stelzneri (A, B–F) and C. tulumbensis sp. nov. (A, G–M), general shell morphology and habitat (P).

(A) Live specimens of C. stelzneri (left) and C. tulumbensis sp. nov. (right). (B) Ventral, (C) lateral, (D) dorsal shell of Clessinia stelzneri, scale bar = five mm. (E) Detail of the aperture, note the inner position of the obstructing teeth and lamellae. (F) Detail of the shell cornet of C. stelzneri with the palatal wall removed to show the undulating lower columellar lamella. (G) Live specimen of C. tulumbensis sp. nov., note axial ribs well marked especially in the body whorl and the lack of periostracal hairs. (H) Ventral, (I) lateral, (J) dorsal shell of the holotype of Clessinia tulumbensis sp. nov. (IBN 883), scale bar = four mm. (K) Ventral, (L) lateral, (M) dorsal shell of a paratype of C. tulumbensis sp. nov. (IBN 571), scale bar = two mm. (N) Shell aperture in C. tulumbensis sp. nov. (O) Detail of the shell cornet of C. tulumbensis sp. nov. with the palatal wall removed to show the deeply undulating lower columellar lamella. (P) Natural microhabitat of Clessinia tulumbensis sp. nov. Photographs by M.G. Cuezzo.

Shell (Figs. 10B–10F and 11A–11D): Fusiform, comprising 8 ½ to 9 slightly convex whorls. Coloration pale to dark brown, sometimes with longitudinal strips clearer in color, other shells with uniform coloration (Figs. 10B–10D). Protoconch with axial, regularly arranged strength ribs, and thin spiral parallel bands delimited by spiral grooves (Fig. 11A). Teleoconch with shallow axial costules separated by regular spaces. Surface of the teleoconch traversed by densely arranged spiral rows bearing two types of periostracal hairs (Figs. 11B–11D). Rows of tall hairs intercalated with three to five spiral rows some of which not bearing hairs while other bearing hairs triangular shaped, less tall, usually in touch with each other through their bases (Figs. 11C and 11D). Ultrastructural ornamentation of body whorl extending over dorsal portion of the cornet. Last portion of the body whorl detached ending into a cornet. Aperture subcircular with five inner teeth and lamellae not connecting to the peristome (Figs. 10B and 10E). Upper columellar lamella long, straight, spirally following the columellar axis. Lower columellar lamella running parallel, slightly undulating, spirally following columellar axis (Fig. 10F). Basal lamella straight, short. Dorsally the body whorl shows a deep groove produced by the basal lamella. Shell measurements represented in Table 1.

Figure 11 Clessinia stelzneri and C. tulumbensis sp. nov., shell ultrastructure.

C. stelzneri: (A) Protoconch and first whorl of the spire, scale bar = 100 μm. (B) Detail of the periostracal ultrastructure of following whorls, scale bar = 1,000 μm. (C) Detail of periostracal hairs, scale bar = 100 μm. (D) Detail of body whorl close to aperture showing periostracal hairs and marked axial costules, scale bar = 100 μm. Clessinia tulumbensis sp.nov.: (E) second and third spire shell whorls with spiral rows without periostracal hairs, scale bar = 300 μm. (F) Contour of body whorl in C. tulumbensis sp. nov. with dense arrange of periostracal spiral rows, scale bar = 100 μm. Photographs by M.G. Cuezzo.

Jaw (Fig. 8B): Markedly horseshoe shaped. Nine plaques with a triangular central one subdivided into three triangular subplaques. Four lateral rectangular shaped plaques at both sides of the central one. Lateral plaques strongly increasing their size toward the tip of the horseshoe. Each plaque traversed by several transversal grooves.

Radula (Figs. 8F–8I): Radular teeth transversally arranged on a straight line. Central tooth tricuspid, with mesocone triangular to rhomboidal. Lateral tooth bicuspids with a high mesocone and a short ectocone in an opposite position to the central tooth. Marginal tooth tricuspid to multicuspids, broader than laterals.

Pallial system: same as in C. cordovana.

Reproductive system (Figs. 12A and 12B): Bursa copulatrix with sac rounded, usually longer than spermoviduct, in some specimens longer than spermoviduct plus albumen gland (Fig. 12A). External limits between epiphallus and penis not evident. Penis cylindrical, long, without penis sheath. Inner morphology of the penis divided into three areas marked by differential pattern of sculpture. Proximal portion globose with higher diameter than restant portions, inner sculpture with pustules. Penial papilla absent. Penis medial sector cylindrical, inner wall with a longitudinal, thick, well-delimited pilaster running from proximal to distal end of the medial zone. Inner sculpture of distal penis portion consisting in three to four longitudinal straight thin pilasters, parallel to each other. Penial sheath absent. Penial retractor muscle short and thick, inserting in penis proximal portion. Epiphallus ⅓ of penial length. Flagellum thinner than epiphallus and ⅔ epiphallus length. Vas deferens thin, running freely along penis, attached to penial retractor muscle (Fig. 12B), then free along epiphallus and inserting between flagellum and epiphallus. Vagina cylindrical, with inner wall with thick, longitudinal pilasters, as long as distal portion of penis.

Figure 12 Clessinia stelzneri (A, B) and Clessinia tulumbensis sp. nov. (C–F), reproductive system.

Clessinia stelzneri: (A) General view, limits penis/epiphallus are indicated, scale bar = two mm. (B) Photograph showing the path of the vas deferens through the proximal penis, epiphallus, then it is attached to the retractor muscle, and later inserts between the epiphallus and flagellum. Clessinia tulumbensis sp. nov.: (C) general view of the reproductive system, scale bar = two mm. (D) Inner sculpture of epiphallus-penis wall, scale bar = three mm. (E) Photograph showing inner sculpture of penis. (F) Relation of the vas deferens and the penis muscular retractor, scale bar = one mm. Abbreviations: ag, albumen gland; bc, bursa copulatrix; e, epiphallus; ec, epiphallus inner constriction; f, flagellum; mr, penial retractor muscle; p, penis; ppi, penial pilaster; ps, penis sheath; s, spermoviduct; v, vagina; vd, vas deferens.

Habitat: calcareous rocky outcrops on mountain slope, under and between roots of woody shrubs.

Distribution (Fig. 5D): Cruz del Eje and Tulumba departments, Córdoba province.

Clessinia tulumbensis sp. nov.

urn:lsid:zoobank.org:act:F565D3BD-03AD-4CC1-8BB1-20D72C5BDF00

Figures 2, 5 and 10–12; Tables 1 and 2.

Odontostomus (Scalarinella) cordovanus striatus Parodiz, 1939: 733;–Breure, 1974: 124.

Clessinia cordovana–Cuezzo, Ovando & Miranda, 2013: 162 [partim].

Type material. Holotype: IBN 883 (preserved in ethanol 96%) (H: 14.9; Dm: 4.61; dm: 3.97; Dap: 3.36; Hap: 4.66). Paratypes: IBS-Ma 311 (3 specimens); IBN 571 (5 specimens, preserved in ethanol 96%); IBN 558 (1 specimen, preserved in ethanol 96%) (H: 17.61; Dm: 3.85; dm: 3.53; Hap: 3.79; Dap: 2.73). Holotype Clessinia cordovana striata, MACN-In 9127.

Type locality. Holotype: Córdoba, Tulumba department, Route 16, between Villa Tulumba and San José de la Dormida (−30.79053, −64.63097; 645 m), October 21, 2017, Cuezzo MG and Dominguez E collectors. Paratypes: IBS-Ma 311, idem to holotype; IBN 571, Córdoba, Tulumba, R 16 (−30.40022, −64.04222; 633 m), November 25, 2018, Cuezzo MG collector; IBN 558, Córdoba, Tulumba, Route 16 before reaching the town of Tulumba (−30.40094, −64.04039; 628 m), November 24, 2008, Cuezzo MG collector.

Etymology. The specific name, “tulumbensis,” is given in reference to Tulumba, the political department of Córdoba Province, Argentina, where the type of the new species was collected.

Definition. Shell with marked wide axial ribs narrowly separated at regular spaces, traversed by major periostracal spiral rows with shallow continuous lamellae. Periostracal hairs absent. Lower columellar lamella in shell aperture deeply undulating. Proximal penis inner sculpture with undulating thin folds in a reticulated disposition and a short thick, medium pilaster. Medial portion with a pilaster of ⅔ the length of penis. Vagina almost the same length of the distal portion of penis.

Description

External features (Figs. 10A and 10G): Soft body, including the sole with dark brownish homogeneous coloration. Foot short with blunt end.

Shell (Figs. 10G–10O and 11E–11F): Turritelliform to subfusiform comprising 9–10 slightly convex whorls (Figs. 10H–10M). Coloration light brown, with whitish ribs. Protoconch with axial strength ribs regularly arranged, with thin spiral parallel bands between ribs (Fig. 11E). Teleoconch with marked oblique, wide axial ribs narrowly separated at regular spaces (Figs. 10H–10M, 11E and 11F). Surface of the teleoconch traversed by major periostracal spiral rows with shallow continuous lamellae, parallel to each other. Between these major spiral rows, two to three minor spiral, undulating shallow lines (Figs. 11E and 11F). Periostracal hairs absent. At ultrastructural level, departing from each major spiral row, growth axial ramified microfolds traversed by spiral lines, giving the appearance of an irregular net (Fig. 11F). Suture deeply impressed. Last portion of body whorl detached from rest of the shell ending into a cornet (Figs. 10I and 10L). Aperture suboval with parietal side slightly excavated (Figs. 10H, 10K and 10N). Peristome simple, thin, slightly expanded and reflexed. A dorsal groove present in upper parietal-palatal side of aperture formed by the detached suture of body whorl. Peristomal ultrastructural sculpture of body whorl extending over dorsal portion of the cornet. Five lamellae present in the interior of the aperture, not connecting to the peristome. Upper columellar lamella long, straight, spirally following the columellar axis. Lower columellar lamella running parallel, deeply undulating, spirally following columellar axis (Fig. 10O). Basal lamella short, located to the right side of the cornet and making a deep indentation on dorsal side of the shell wall (Figs. 10J and 10M). Umbilicus narrow. Shell measurements represented in Table 2.

Jaw (Fig. 8C): Horseshoe shaped, less open than in C. cordovana. A total of 12 plaques with a triangular central one subdivided into three subplaques. Six lateral narrow rectangular shaped plaques at both sides of the central one. Lateral plaques slightly increasing their size toward the tip of the horseshoe. Each plaque traversed by several thin transversal grooves.

Pallial system: idem to C. cordovana.

Reproductive system (Figs. 12C–12F): Ovotestis embedded into digestive gland into the fourth or fifth spire whorls. Hermaphroditic duct inserting at distal portion of the albumen gland. Seminal receptacle swollen. Fertilization pouch-spermathecal complex long, digitiform broaden at its base. Albumen gland spread within the sixth whorl. Bursa copulatrix with sac rounded and folded over proximal section of the duct (Fig. 12C). Bursa copulatrix duct, longer than spermoviduct, surrounding the spermoviduct and running toward basal portion of the albumen gland. External limits between epiphallus and penis not evident, only differentiated by its inner sculpture. Penis cylindrical, long, with a short, thin penis sheath overlapping distal penis (Figs. 12C and 12D). Inner morphology of the penis divided into three areas marked by differential pattern of sculpture. Proximal portion slightly swollen than remaining portions, inner sculpture with undulating thin folds in a reticulated disposition and a short thick, medium pilaster (Figs. 12D and 12E). Penial papilla absent. Penis medial portion long, cylindrical, inner wall with rhomboidal to hexagonal pustules on the surface, with a pilaster of ⅔ the length of the medial penis portion. Distal penis short, thinner than medial portion with inner sculpture consisting in three to four longitudinal, straight, thin pilasters, parallel to each other. Penial retractor muscle thin and long, inserting in penis proximal portion (Fig. 12F). Epiphallus ¼ of penial length, with an inner constriction cutting the longitudinal, thin folds. Flagellum thinner than epiphallus and ½ epiphallus length. Vas deferens thin, running freely along penis, attached to penial retractor muscle, then free along epiphallus and inserting between flagellum and epiphallus (Fig. 12F). Vagina cylindrical, with a distal portion thinner in diameter than the proximal, inner wall with longitudinal pilasters. Vagina almost the same length of the distal portion of penis. Atrium short.

Habitat (Fig. 10P): Living in rocky outcrops, on and under shrubs. Always dry environments.

Distribution (Fig. 5D): This species has a small area of distribution located in northwestern Córdoba within Tulumba, and Totoral departments, Argentina. Western limit for C. tulumbensis sp. nov. area of distribution is in Sierra de Macha, extending to the east toward Villa Tulumba, and to the north toward Cerro Colorado. Localities of occurrences are all below 700 m, in Chaco Serrano subecoregion.

DNA sequence data. Partial sequences of mitochondrial COI and 16S-rRNA genes, and the nuclear ITS-2 region from two paratypes (IBN 883, specimens 1 and 2) have been deposited in GenBank with accession numbers: MG963436 and MG963437 for COI; MG963462 and MG963463 for 16S-rRNA, and MH789460 and MH789461 for ITS-2.

Remarks. The new species, Clessinia tulumbensis sp. nov. include Clessinia cordovana striata (Parodiz, 1939). The name striata has not been used here to avoid homonymy with Pupa striata Spix, 1827, the type species of Spixia, since in the present study the genera Clessinia and Spixia are proposed as synonymous. The new species with its own holotype and paratypes is defined based on live-collected material from which DNA sequences were obtained and the anatomy described. In this sense, although the Parodiz name is preoccupied, we are not replacing the name proposed by him in 1939 but creating a new species with its own type series. C. tulumbensis sp. nov. has clear differences with the remaining species of the cordovana-group showing shell axial ribs more marked than in C. cordovana and C. stelzneri. The periostracum lacks spines or hairs, only major periostracal spiral rows with shallow continuous lamellae are present. C. tulumbensis sp. nov. has a clear delimited pilaster in the medial penis portion, also present in C. stelzneri, but absent in C. cordovana. The penis shows a short, thin penial sheath overlapping part of the penial distal portion and the vagina is short as in C. stelzneri. C. tulumbensis sp. nov. has a narrow distribution area in northern Córdoba, occurring in sympatry with C. stelzneri in some localities within eastern portion of Tulumba department.

Clessinia pagoda Hylton Scott, 1967

Figures 2, 8 and 13–15.

Clessinia pagoda Hylton Scott, 1967: 98;–Fernández, 1973: 144;–Breure, 1974: 120;–Neubert & Janssen, 2004: 221, pl. 19, fig. 247;–Cuezzo, Miranda & Ovando, 2013: 163, fig. 2A.

Type material. Holotype not found; paratypes MACN-In 27284 (15 specimens, type locality), Córdoba, Quilpo, 2-3/IV/1967. Cichero-Biraben Leg. Hylton Scott det., H: 22.2–19.8 (mean = 22.5); Hap: 7.4–6.6 (mean = 7.04); Dap: 5.9–4.4 (mean = 5); Dm: 8.4–6.5 (mean = 7.3); IFML-Moll 14239, Córdoba, Quilpo, 5/4/1967, Birabén-Cichero leg.; MLP-Ma 11077, Córdoba, Quilpo (paratype); SMF 220916/2a.

Type locality. Argentina, Córdoba, Cruz del Eje department, Sierra Chica de Córdoba, Quilpo.

Description

External features (Figs. 13A and 13B): Animal dark brown to black, with light brown ocular tentacles. Homogeneous coloration over the cephalopedial region, same specimens with sole light cream. Usually shells of live snails covered with a coat of sand granules (Fig. 13A).

Figure 13 Clessinia pagoda, general shell morphology and habitat.

(A) Live specimen with a thick layer of sand granules covering periostracal sculpture. (B) Live specimen partially cleaned, crawling on granitic rock. (C–E) Paratype (MACN-In 27284) eroded shell without periostracal complex sculpture on ventral (C), lateral (D) and dorsal (E) views. (F–H) Ventral (F), lateral (G) and dorsal (H) views of a cleaned shell showing periostracal structures typical of the species, scale bar = five mm. (I, J) Shape variability of the aperture, scale bar = five mm. (K) View of the habitat in San Marcos Sierras on the road to Quilpo, type locality. (L) Typical microhabitat of C. pagoda. Photographs by M.G. Cuezzo.

Shell (Figs. 13C–13J and 14A–14F): Subpyriform with conic spire, solid. Seven to eight whorls with median keel in each spire whorl (Figs. 13C–13E). Body whorl with convex contour. Homogeneous light brown when periostracum is present (Figs. 13A, 13B and 13F–13H). Protoconch with axial strength ribs regularly arranged, space between axial ribs with thin spiral parallel bands (Fig. 14A). Teleoconch with oblique shallow ribs slightly marked (Fig. 14B). Each spire whorl with an equatorial and/or lower spiral row bearing rounded lamella of 300–400 μm tall and 200–250 μm wide (Figs. 13F–13H, 14B, 14C and 14E). Each lamella superimposed with the following in a row (Fig. 14C). Several major and minor spiral rows parallel to the lamellae medial row. Suture between whorls also with a row of lamellae (Fig. 14E). Lamellae are lost in abraded specimens without periostracum (Figs. 13C–13E). Minor, shallower spiral rows regularly spaced between major rows with lamellae. Space between minor rows showing microaxial folds with the appearance of an irregular net (Fig. 14D). Sculpture of body whorl consisting in at least five major spiral rows of smaller lamellae than former described for spire. Variable number of minor spiral rows between major rows of lamellae (Fig. 14E). Aperture detached from body whorl forming a cornet, peristome expanded (Figs. 13D and 13G). Microsculpture of body whorl prolonged dorsally over peristome (Fig. 14F). Suture of body whorl when detached forming a marked keel that produce a marked dorsal angle of the aperture (Figs. 13C and 13D). Aperture rounded to oval (Figs. 13I and 13J). Dorsally, last portion of body whorl with a marked groove (Fig. 14F). Five lamellae obliterating the aperture. Umbilicus narrow.

Figure 14 Clessinia pagoda, shell ultrastructure.

(A) Protoconch and following whorls of the spire, note the change of periostracal sculpture between protoconch and other whorls, scale bar = 100 μm. (B) Fourth and fifth shell whorls, note the spire line bearing lamellae located in the low portion of each teleoconch whorl, scale bar = 100 μm. (C) Lamellae superimposed with the following in a spiral row in the shell whorl, scale bar = 10 μm. (D) Detail of microaxial folds with the appearance of an irregular net in the space between minor rows, scale bar = 10 μm. (E) Lateral view of a teleoconch whorl with spiral major and minor rows, and the row bearing lamellas, scale bar = 200 μm. (F) Dorsal view of the body whorl showing the microsculpture prolonged over the expanded peristome, scale bar = 1,000 μm. Photographs by M.G. Cuezzo.

Jaw (Fig. 8D): Horseshoe shaped. Eleven plaques with a triangular central one subdivided into three longitudinal subplaques. Lateral plaques quadrangular to rectangular shaped, increasing their size toward the tip of the horseshoe. Each plaque traversed by several transversal grooves.

Pallial System (Figs. 15A and 15B): Pulmonary roof thin, traversed by few veins mostly concentrated on distal portion. Kidney triangular, short, ¼ the length of the pulmonary roof. Kidney with several longitudinal folds in its interior (Fig. 15B). Secondary ureter closed over most of its length, opening slightly before rectum. Pallial gland thin, parallel to mantle collar. Afferent vein parallel to main pulmonary vein. Mantle collar deeply marked by shell lamellae.

Figure 15 Clessinia pagoda, anatomy.

(A) General view of the ventral side of the pulmonary cavity, scale bar = five mm. (B) Detail of the kidney, scale bar = five mm. (C) General view of the reproductive system, limits penis/epiphallus and epiphallus/flagellum are indicated, scale bar = five mm. (D) Detail of the fertilization pouch-spermathecal complex, scale bar = two mm. (E) External view of the phallic complex, scale bar = five mm. (F) Inner view showing the sculpture of epiphallus-penis wall, scale bar = one mm. Abbreviations: ag, albumen gland; bc, bursa copulatrix; e, epiphallus; ec, epiphallus inner constriction; f, flagellum; fp, fertilization pouch-spermathecal complex; hd, hermaphroditic duct; k, kidney; mc, mantle collar; p, penis; pc, pericardium; pf, pulmonary fold; ps, penis sheath; r, rectum; s, spermoviduct; v, vagina.

Reproductive system (Figs. 15C–15F): Ovotestis formed by a bunch of digitiform, long acini embedded in the digestive gland. Hermaphroditic duct inserting at distal portion of the albumen gland, with its medial portion, corresponding to the seminal vesicle, swollen. Fertilization pouch-spermathecal complex long, digitiform broaden at its base (Figs. 15C and 15D). Bursa copulatrix with sac rounded and its duct slightly swollen at its base. Bursa copulatrix sac level with distal portion of albumen gland, its duct surrounding the spermoviduct, longer than spermoviduct in total length (Fig. 15C). Phallic complex formed by flagellum, epiphallus, and penis. External limits between epiphallus and penis not evident, only differentiated by its inner sculpture. Flagellum tapering toward its tip, thinner than epiphallus and about as long as epiphallus length (Fig. 15E). Epiphallus about the same length than penis, slightly increasing its diameter toward distal portion, with an inner constriction cutting the inner surface into two portions (Fig. 15E). Proximal portion with thick, pronounced, longitudinal pilasters, while distal portion with thin, scatter folds more separated between each other. Penis cylindrical, long, with a short, thin penis sheath overlapping its distal portion (Figs. 15E and 15F). Penis also separated from epiphallus by a thin inner constriction. Inner surface of penis wall divided into three areas marked by differential pattern of sculpture. Proximal portion externally slightly swollen than resting portions, with inner sculpture formed by tightly appressed thin folds arranged in a reticular shape. Penial papilla absent. Penis medial portion long, cylindrical, inner wall traversed by thin folds that toward distal penis became diagonally arranged (Figs. 15E and 15F). Distal penis short, thinner than medial portion with inner sculpture consisting in three to six longitudinal, straight, thin folds, parallel to each other. Penial retractor muscle thick and short inserting in penis proximal portion. Vas deferens thin, running under penis sheath and then freely along penis, attached to penial retractor muscle, then free along epiphallus and inserting between flagellum and epiphallus. Vagina cylindrical, even in diameter, inner wall smooth or with shallow longitudinal pilasters. Vagina longer than distal portion of penis (Fig. 15C). Atrium short.

Habitat. Found in mountains with xerophytic vegetation usually under rocks or in crevices in rocks.

Distribution (Fig. 5D): Clessinia pagoda is only known from the localities of Quilpo and San Marcos Sierras in Córdoba province, Cruz del Eje department, in the Chaco Serrano. C. pagoda is a narrow range endemic species from northwestern Córdoba. The Cerro de la Cruz is close to Quilpo and to San Marcos Sierras, mountain where the species is easily found from 600 to 900 m of altitude.

Remarks. Clessinia pagoda paratypes specimens are completely worn out (Figs. 13C–13E) and therefore all the miscrosculpture of the periostracum is lost. Living snails are found under clay or granite rocks. They are usually camouflaged with sand grains of the substrate that adhere to the periostracum (Fig. 13A), but the fragile lamellae and complex structures of the periostracum are not perceived until the shell is clean (Figs. 13F–13H). Strikingly, other carinated, rare land snail species, Plagiodontes weyenberghii (Doering, 1875 [1877a]) of the family Odontostomidae (Pizá & Cazzaniga, 2012) also occurred in scatter areas through Córdoba in the Chaco Serrano subecoregion.

Clessinia nattkemperi (Parodiz, 1944)

Figures 2, 8 and 16–18.

Cyclodontina (Scalarinella) nattkemperi–Parodiz, 1944: 1–2, figs. A-D;–Parodiz, 1957: 29;–Breure, 1974: 119.

Clessinia nattkemperi–Fernández, 1973: 144;–Cuezzo, Miranda & Ovando, 2013: 163.

Type material. Holotype MACN-In 25713; Paratypes MACN-In 25713-1 (13 shells, type locality) (Tablado & Mantinian, 2004).

Type locality. Argentina, Catamarca, Esquiú department, Pomancillo, 23 km from San Fernando del Valle de Catamarca, F. Nattkemper leg. July 1943.

Description

External features (Figs. 16A and 16B): Animal pale brown, homogeneous coloration with ocular tentacles of the same color. A dorsal row of pustules from mantle collar ending between the two ommatophoral tentacles. Foot short with blunt extreme.

Figure 16 Clessinia nattkemperi, general shell morphology and habitat.

(A and B) Live specimens of Clessinia nattkemperi. (C) Ventral, (D) lateral and (E) dorsal views of the Holotype specimen (MACN-In 25713), note the eroded shell surface lacking all the periostracal ornamentation. (F) Ventral, (G) lateral and (H) dorsal views of a shell with periostracal ornamentation, scale bar = three mm. (I–K) Detail of the teeth in a suboval (I), subquadrate (J), and narrower suboval (K) shell aperture, scale bar = two mm. (L) View of the Chaco Serrano habitat of Clessinia nattkemperi in Catamarca. (M) View of the microhabitat of C. nattkemperi. (N) Live specimens between spines in cactuses. Photographs by M.G. Cuezzo.

Shell (Figs. 16C–16K and 17): Fusiform with gradual increase of diameter toward body whorl. Eight shell whorls with convex contour. Shell yellowish to pale golden. Holotype completely worn out with no periostracal ornaments (Figs. 16C–16E). Protoconch consisting of the first two whorls, with strength axial ribs, space between axial ribs with thin spiral parallel bands. Teleoconch with axial ribs and conspicuous periostracum ornamentation when present (Figs. 16F–16H). Periostracal sculpture consists of 10–20 thin spiral rows bearing triangular spines (Figs. 17A–17E) separated at regular spaces. Spines with wide base (50–60 μm) and about 100 μm tall (Figs. 17B and 17C). Space between spiral rows traversed by axial irregular microfolds cut by spiral or diagonal microribs forming an irregular net. Only the major type of spiral row is present, some of them without spines intercalated with the ones bearing spines (Fig. 17B). Aperture slightly detached from body whorl forming a shallow cornet (Figs. 16G, 17D and 17E), subovate to subquadrate, with a marked dorsal groove in upper portion of the aperture. Peristome expanded. Microsculpture of body whorl prolonged dorsally over peristome (Fig. 17D). Five inner teeth or lamellae not touching the peristome (Figs. 16I–16K). Upper palatal tooth triangular, not present in some specimens (Fig. 16J). Lower columellar lamella with rounded or rectangular shape when viewing from outside (Figs. 16I–16K).

Figure 17 Clessinia nattkemperi, shell ultrastructure.

(A) Second and third whorls of the teleoconch showing general aspect of the sculpture, scale bar = 200 μm. (B) Spiral rows bearing triangular spines, scale bar = 100 μm. (C) Detail of the spines triangular shaped with wide base, scale bar = 10 μm. (D) Dorsal view of the body whorl at the level of the aperture showing microsculpture prolonged over peristome, scale bar = 1,000 μm. (E) Lateral view of the body whorl with the detached aperture, scale bar = 1,000 μm. Photographs by M.G. Cuezzo.

Jaw (Fig. 8E): Wide horseshoe shaped formed by 15 plaques, medial one triangular in shape and subdivided into three narrower plaques. Lateral plaques very narrow and about same size. The three last plaques on each side of the jaw broader than more central plaques.

Pallial system: Pulmonary roof thin and long traversed by few veins mostly concentrated on distal portion. Kidney triangular, short, a quarter of the total length of the pulmonary roof. Secondary ureter closed over most of its length, opening slightly before rectum. Pallial gland thin, parallel to mantle collar. Afferent vein parallel to main pulmonary vein.

Reproductive system (Figs. 18A–18D): Ovotestis formed by a bunch of digitiform, long acini embedded in the digestive gland. Hermaphroditic duct inserting at distal portion of the albumen gland, with a swollen medial portion, corresponding to the seminal vesicle (Fig. 18A). Fertilization pouch-spermathecal complex long, thin, digitiform, broaden at its base. Bursa copulatrix with sac rounded with its duct even in diameter along its length. Bursa copulatrix sac level with distal portion of albumen gland, its duct surrounding the spermoviduct, longer than spermoviduct in total length. Phallic complex formed by flagellum, epiphallus, and penis. External limits between epiphallus and penis not evident, only differentiated by its inner sculpture. Flagellum thin, tapering toward its tip, thinner than epiphallus and shorter in length (Figs. 18A and 18B). Epiphallus slightly shorter than penis, increasing its diameter toward distal portion, with an inner constriction cutting the inner surface into two portions. Proximal portion with thin, straight longitudinal pilasters, distal portion shorter with longitudinal folds less separated between each other and scalloped outline. Both portions internally separated by an inner constriction (Fig. 18C). A cylindrical papilla of the epiphallus with distal digitiform extensions is present. Penis mostly cylindrical, long, with a short, thin, transparent penis sheath overlapping its distal portion. Proximal portion more swollen than remaining portions, globular in some specimens, with inner sculpture formed by tightly appressed thin folds arranged in a reticular shape with a central, short pilaster (Figs. 18B and 18D). Penial papilla absent. Penis medial portion long, cylindrical, inner wall with thin, parallel folds, with marked festooned outline (Fig. 18D). Distal penis short, thinner than medial portion with inner sculpture consisting in three to six longitudinal, straight, thin folds, parallel to each other. Penial retractor muscle thick and short inserting in penis proximal portion (Fig. 18A). Vas deferens thin, running under penis sheath and then freely along penis, attached to penial retractor muscle, then running parallel and attached to epiphallus by thin tissue and inserting between flagellum and epiphallus. Vagina cylindrical, short, even in diameter, inner wall smooth, or with shallow longitudinal pilasters. Vagina shorter than distal portion of penis. Atrium short.

Figure 18 Clessinia nattkemperi, anatomy.

(A) General view of the reproductive system, scale bar = five mm. (B) Exterior view of the phallic complex, limits penis/epiphallus and epiphallus/flagellum are indicated, scale bar = five mm. (C) Detail of inner wall sculpture of the epiphallus, scale bar = one mm. (D) Detail of the inner sculpture of the penis wall, scale bar = one mm. Abbreviations: ag, albumen gland; bc, bursa copulatrix; dp, penis distal portion; e, epiphallus; ef, epiphallic folds; ep, epiphallic papilla; f, flagellum; hd, hermaphroditic duct; mp, penis medial portion; mr, penis retractor muscle; p, penis, pp, proximal penis; ps, penis sheath; s, spermoviduct.

Habitat (Figs. 16L–16N): Clessinia nattkemperi is found in close association with xerophytic plants, mainly cactuses, in patches of Chaco Serrano subecoregion. Specimens usually are hiding between long spines of cactuses or below dead cactuses branches lying over the ground. Found in sandy, dry substrate. Not found in rock crevices as other species of the genus.

Distribution (Fig. 5B). This species is endemic to the Sierra de Graciana mountain system and was only collected around the locality of Pomancillo, in Catamarca province, Northwestern Argentina. Dry Chaco ecoregion, Chaco Serrano subecoregion.

Remarks. C. nattkemperi is the species of the genus more similar to the former Spixia in general shell shape morphology. Its shell aperture is only slightly detached from the body whorl. It also shows triangular periostracal lamellae that are similar to the ones present in C. martensii (Doering, 1874 [1875]) and C. tucumanensis (Parodiz, 1941).

Discussion

Traditional used characters for taxonomic diagnosis in Odontostomidae, such as the shell morphology, provide important information for species identification, but due to their intraspecific variability they should be carefully considered. When Doering described Clessinia in 1875, he mentioned that the number of plaques of the jaw would be the best character to differentiate the new created genus from Odontostomus, Plagiodontes, and Spixia. However, the number of plaques in Clessinia overlaps with the ones present in species of other genera, so by itself this character is not enough for genera differentiation.

Our combined morphological and molecular study allows us to propose that the so called cordovana-group is formed by three species, C. cordovana, C. stelzneri, and C. tulumbensis sp. nov. The shell in C. tulumbensis sp. nov. has prominent, well-marked ribs, more raised than in C. cordovana and C. stelzneri. C. cordovana and C. tulumbensis sp. nov. have thinner shells than the other species. Among the cordovana species complex, the shape of the lower columellar lamella is deeply undulating around the columellar axis in C. tulumbensis sp. nov. and straighter in C. cordovana and C. stelzneri, thus showing to be a good character for species identification. The periostracum is of special taxonomic interest because it bears distinct microscale architectures. However, where and how these structures are formed is yet unknown for the majority of the species (Allgaier, 2011). Haired shells occur in several species of Stylommatophora, as for example, in families Polygyridae, Helicidae, Hygromiidae, Clausiliidae, Vertiginidae, and Solaropsidae. These families are distantly related suggesting that these features have evolved several times independently. In some cases, the ornamentation can be a response to structural demands from the environment, including camouflage and defense against predators and parasites. These ornaments can be the support for sand granules, dirt, or any other component of the natural habitat that can camouflage the shell. Pfenninger et al. (2005) suggest that hairs on shells of Trochulus Chemnitz, 1786 confer a selective advantage in humid habitats only and that are lost in drier habitats. Strikingly, in Argentina drier habitats hold the species with longer hairs such as C. cordovana from central western mountains in Córdoba. Clessinia pagoda is frequently found covered by a thick layer of sand grains overlapping the shell spiral rows but not completely the lamellae. The coloring of the shells covered by sand is perfectly camouflaged with the rocks where they live. In Clessinia, the type and diversity of periostracal ornamentation needs to be further investigated taking into account not only their ecological niche differentiation but also species reproductive behavior. Anatomical studies show that the reproductive system holds the most useful characters for taxonomic identification as in other stylommatophoran snails. Inner sculpture of the penis in the five species studied show particular characters. The presence of a penial pilaster and a sheath also contributes to species differentiation in the cordovana-species group.

The geometric morphometric analyses performed in this study confirmed the distinctiveness among all the species here treated. While C. pagoda and C. nattkemperi are clearly different, there is some degree of overlap in shell shape among the species of the cordovana-group. While C. cordovana has a slim body whorl with a suboval aperture, C. stelzneri has a body whorl more expanded and voluminous with subcircular aperture and C. tulumbensis sp. nov. has an intermediate shape of body whorl with expansion of the central portion of the aperture. When the cordovana-species complex was analyzed alone, shell differences are more evident but still a degree of overlap exists. Clessinia cordovana from Cerro del la Cruz and surrounding San Marcos Sierras areas shows the typical shapes according to the species description, while specimens from Sierra de Pocho are more closely related in shape to C. stelzneri. In the case of C. tulumbensis sp. nov., Cerro Colorado is the locality that shows specimens with smaller aperture and spire first whorls more expanded. Even when geometric morphometrics is useful for taxonomic identification, we support the necessity of a comparative analysis using also anatomical studies for a correct taxonomic identification of specimens within the cordovana-group.

Molecular studies performed with the nuclear marker, revealed that the ITS-2 region contains much less polymorphism than the mitochondrial genes and does not exhibit enough genetic variation to determine species relationships among Clessinia and Spixia species, which are shown as a polytomy in our phylogenetic reconstructions. These results conform well to Breure & Romero (2012), who obtained poor resolution at lower taxonomical levels within phylogenetic trees of Orthaliciodea by using the same nuclear region. Because this nuclear region seems to be too conserved to depict specific relationships, further studies with more nuclear markers are required in order to reconstruct fully resolved phylogenetic trees in the genus Clessinia. On the other hand, our analyses performed with 16S-rRNA showed that C. cordovana differs from C. stelzneri by genetic divergences ranging between 17% (p distance) and 19.3% (K2P distance) and from C. tulumbensis sp. nov. by distances ranging between 17% (p distance) and 19.8% (K2P distance), thus suggesting that C. cordovana is a different species from C. stelzneri and C. tulumbensis sp. nov. Regarding pairwise interspecific divergence between C. stelzneri and C. tulumbensis sp. nov., the greatest genetic distances we found are less than 2% (1.1–1.8%). Similarly, based on the COI locus we found that C. cordovana differs from C. stelzneri by genetic distances ranging between 14.3% (p distance) and 17% (K2P distance), and from C. tulumbensis sp. nov. by divergences ranging between 14.9% (p distance) and 19% (K2P distance), while distances between C. stelzneri and C. tulumbensis sp. nov. ranged between 1.3% and 4.1%. As with the ribosomal marker, these COI-based values suggest that C. cordovana represent a different species from C. stelzneri and C. tulumbensis sp. nov.

Our analyses with two different methods to test if morphological species of Clessinia satisfied the criteria to be considered different evolutionary genetic species, allowed us to recognize C. cordovana, C. nattkemperi, and C. pagoda as distinct evolutionary genetic species. However, the results from ABGD and K/θ approaches failed to recognize the morphological diversity between C. stelzneri and C. tulumbensis sp. nov., and the phylogenetic trees did not support the presence of separate species for both morphological species. The trees showed C. stelzneri as paraphyletic with respect to C. tulumbensis sp. nov., with DNA sequences of C. tulumbensis sp. nov. nested among those of C. stelzneri. Furthermore, the distance values obtained between C. stelzneri and C. tulumbensis sp. nov. are relatively low. This low genetic divergence suggest that the two species have evolved relatively recently, with C. tulumbensis sp. nov. having evolved within C. stelzneri, and would explain the reduced resolution of species boundaries detection using genetic data, as both ABGD and K/θ approaches are known to fail in cases of very recent speciation (Puillandre et al., 2012; Birky, 2013). Thus, based on the molecular markers analyzed, is not possible to assess to which extent intrinsic factors (e.g., hybridization, introgression) or adaptive evolution are implicated in speciation, and further research involving more populations across the distribution area of both species, crossbreeding experiments, and more suitable molecular markers accounting for low divergence times are needed to testing hypotheses of speciation on these Clessinia species. In addition, when the COI sequences of Breure & Romero (2012) were included together with the sequences obtained here, some interesting situations occur. Based only on the partial COI gene, the phylogenetic trees of Clessinia and Spixia species again showed C. stelzneri as paraphyletic when including the sequences provided by Breure & Romero (2012). At this point, we revised the taxonomic identity of the material from IML BD575 currently deposited at IBN, on which Breure & Romero (2012) based their findings and found that this material corresponds in morphology to C. tulumbensis sp. nov. and not to C. cordovana, as stated by these authors. In addition, we observed that S. popana was included within the C. stelzneri plus C. tulumbensis sp. nov. group, a finding which needs further research based on new material of this Spixia species in order to rule out a misidentification.

Clessinia gracilis was created by Hylton Scott in 1966 on the base of a single shell from the locality of “La Puerta,” Ambato, in Catamarca, Argentina. Hylton Scott (1966) mentioned that the shell of C. gracilis had “teeth toward the interior of the opening while in the other species they are mounted on the reflection of the lip.” This description of the shell aperture can, however, be referred to the aperture morphology of all the species of the genus, where the position of the teeth is always located in the interior of the shell aperture. The holotype of C. gracilis [MACN 6421] has a distinctive arrange of the teleoconch axial ribs, thick and more separated than the ones present in both C. cordovana and C. tulumbensis sp. nov. During successive fieldwork carried out the past last years, no specimen corresponding to C. gracilis was found in or around the type locality in Catamarca. Clessinia gracilis had been previously synonymized with C. cordovana due to its similar general shape and size (Cuezzo, Miranda & Ovando, 2013). Nonetheless, Breure & Romero (2012) treated C. gracilis and C. cordovana as different species, but the species identification of the material used in that molecular study raised our suspicions because its locality (Córdoba: Quilpo) is where C. cordovana is very abundant. Moreover, Quilpo is distant by a few kilometers from San Marcos Sierras where we have found specimens with more marked axial ribs. Based on COI sequence, C. gracilis groups together with C. cordovana in our phylogenetic trees, with genetic distances ranging between 7.1% (p distance) and 8.9% (K2P distance) from our C. cordovana specimens, and recognized as a different species by the ABGD analysis. Therefore, the identity of what Breure & Romero (2012) consider as C. gracilis should be tested using morphological information to confirm the correct taxonomic identification, whose morphological differentiation might imply the recognition of C. gracilis as a valid species. Regarding C. nattkemperi and C. pagoda, we found genetic distances greater than 10% in relation to the other Clessinia species, and both are recognized as different genetic species according to ABGD and K/θ approaches. However, a finding in need of further investigation is the close relationship we obtained in our phylogenetic analyses between C. pagoda and S. cuezzoae, which exhibits very different morphologic features (Salas Oroño, 2010), but whose genetic distance was about 3.5% for the 16S-rRNA locus and ranged from 4.8% (p distance) to 5.4% for the COI marker.

The results obtained here have taxonomic implications at the genus level for Clessinia and Spixia. On the basis of morphological evidence Spixia and Clessinia are easily externally distinguished because former Clessinia species have the aperture detached from the body whorl forming a cornet, periostracal microsculpture extended over dorsal portion of the peristome, presence of five inner teeth on the shell aperture instead of the three–four in Spixia. However, there are several other morphological similarities among species of both genera in general shell shape, type of periostracum microsculpture, reproductive anatomy, presence of a pallial gland, besides the overlap in geographic ranges. Molecular evidence suggest that taxa placed in former genera Spixia and Clessinia belong to the same group, and both genera as currently known until now would be paraphyletic (Breure & Romero, 2012). Molecular results obtained here support this previous finding. Nonetheless, previous phylogenies based on molecular data, only included few representatives’ species of the genus Spixia. Moreover, its type species S. striata (Spix, 1827), is still only known by its shell description with no information on its internal anatomy and molecular identity. Nonetheless, based on the available evidence, Clessinia and Spixia are synonymous, and according to the principle of priority (ICZN Code, Art.23.1) the valid name of the taxon should be Clessinia Doering, 1875 which has priority over Spixia Pilsbry & Vanatta, 1894.

Conclusions

Studied species of Clessinia are endemic to the Chaco Serrano subecoregion, restricted to small patches between the Semiarid Chaco and Yungas ecoregion in north-central Argentina.

The shell in Clessinia provided useful information for species identification. The aperture, with a continuous peristome, is detached from the body whorl. The apertural teeth are five, always located in the interior of the cornet mostly with no relation to the peristome. The periostracum, a layer that produces a conspicuous ornamentation in Clessinia species has taxonomic importance for the identification of the species. Each species has a particular periostracal structure of hairs of different length and densities and lamellas of different shape and sizes.

Morphological and molecular evidence support the composition of Clessinia cordovana-complex by three species, Clessinia cordovana, Clessinia stelzneri, and Clessinia tulumbensis sp. nov. Shell geometric morphometrics support our results on the species composition of the cordovana-complex but cannot be used alone as a single source of information for species identification.

The genitalia and jaw are also useful sources of characters for species identification. Inner penial sculpture, comparative length of flagellum, proportion of epiphallus respect to penis, the attachment of the vas deferens to a thick penis muscular retractor are the most conspicuous characters.

Based on our current analyses, and on previous findings (Breure & Romero, 2012) Clessinia and Spixia are synonymous, and according to the principle of priority (ICZN Code, Art.23.1) the valid name of the taxon should be Clessinia Doering, 1875 which has priority over Spixia Pilsbry & Vanatta, 1894.

Supplemental Information

Supplemental Information 1 Analysis using landmarks in shell lateral view of all the Clessinia species treated.

Coordinates of the landmarks in two dimensions, digitized with Tpsdig 2, selected in lateral view of the shell of the species of Clessinia (C. cordovana, C. stelzneri, C. tulumbensis, C. nattkemperi and C. pagoda).

Click here for additional data file.

Supplemental Information 2 Bayesian tree of Clessinia and Spixia species based on the nuclear 5.8S-ITS2-28S region.

The posterior-probability values for BI and bootstrap values for the ML tree are shown above the branches. Numbers within groups are GenBank accession numbers.

Click here for additional data file.

Supplemental Information 3 Additional material studied from different collections.

Material from different malacological collections studied. Abbreviations: IBN, Instituto de Biodiversidad Neotropical, Tucumán, Argentina; IFML-Moll, Instituto-Fundación Miguel Lillo, Tucumán, Argentina; MACN-In, Museo Argentino de Ciencias Naturales, Buenos Aires, Argentina.

Click here for additional data file.

Supplemental Information 4 16S Sequences.

Click here for additional data file.

Supplemental Information 5 COI Sequences.

Click here for additional data file.

Supplemental Information 6 Landmarks in lateral view.

Coordinates of the 7 Landmarks in two dimensions, digitized with Tpsdig 2, selected in side view of the shell of all the Clessinia species considered in this study. Coordinates used in the Analysis of Canonical Variation (CVA) in MorphoJ.

Click here for additional data file.

Supplemental Information 7 Landmarks in ventral view.

Coordinates of the 14 Landmarks in two dimensions, digitized with Tpsdig 2, selected in ventral view of the shell of all the Clessinia species considered in this study. Coordinates used in the Analysis of Canonical Variation (CVA) in MorphoJ.

Click here for additional data file.

Supplemental Information 8 Landmarks of the cordovana-species complex.

Coordinates of the 14 Landmarks in two dimensions, digitized with Tpsdig 2, selected in ventral view of the shell of the species of Clessinia cordovana species complex (C. cordovana, C. stelzneri and C. tulumbensis), from different localities. Coordinates used in the Analysis of Canonical Variation (CVA) in MorphoJ.

Click here for additional data file.

Maria Gabriela Cuezzo would like to thank E. Dominguez and D. Dos Santos for companion and support in collecting land snails during field work. Thanks are also extended to the staff of the Centro Integral de Microscopia Electronica (CIME) for help in preparation of specimens and use of the SEM and to L. Cristobal for her assistance with GIS analysis. We also would like to thank the detailed suggestions and corrections made by Gary Rosenberg and an anonymous reviewer as well as the editor, Rudiger Bieler that improved our manuscript.

Additional Information and Declarations

Competing Interests

Author Contributions

Field Study Permissions

DNA Deposition

Data Availability

New Species Registration

The authors declare that they have no competing interests.

Maria Gabriela Cuezzo conceived and designed the experiments, performed the experiments, analyzed the data, contributed reagents/materials/analysis tools, prepared figures and/or tables, authored or reviewed drafts of the paper, approved the final draft, wrote de paper.

Maria Jose Miranda conceived and designed the experiments, performed the experiments, analyzed the data, contributed reagents/materials/analysis tools, prepared figures and/or tables, authored or reviewed drafts of the paper, approved the final draft, wrote the paper.

Roberto Eugenio Vogler conceived and designed the experiments, performed the experiments, analyzed the data, contributed reagents/materials/analysis tools, prepared figures and/or tables, authored or reviewed drafts of the paper, approved the final draft, wrote the paper.

Ariel Anibal Beltramino conceived and designed the experiments, performed the experiments, analyzed the data, contributed reagents/materials/analysis tools, prepared figures and/or tables, authored or reviewed drafts of the paper, approved the final draft, wrote the paper.

The following information was supplied relating to field study approvals (i.e., approving body and any reference numbers):

Field and collection permits for this study were issued by National Parks Administration (DRNOA 126/17) and Environment and sustainable development secretary, Catamarca province (SEAyDS 208/16).

The following information was supplied regarding the deposition of DNA sequences:

GenBank accession numbers: MG963434–MG963464 and GenBank MH789452–MH789466.

The following information was supplied regarding data availability:

The raw data is available in Supplemental Files.

The following information was supplied regarding the registration of a newly described species:

Publication LSID: urn:lsid:zoobank.org:pub:8DB0CC34-AE26-44BA-B7F8-A5F17254BD13.

Clessinia tulumbensis sp. nov.

urn:lsid:zoobank.org:act:F565D3BD-03AD-4CC1-8BB1-20D72C5BDF00.

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
