# Peer review of "From morphology to molecules: a combined source approach to untangle the taxonomy of Clessinia (Gastropoda, Odontostomidae), endemic land snails from the Dry Chaco ecoregion"

_PeerJ, doi:10.7717/peerj.5986_

## Round 0.1 · original submission · Major Revisions

· Academic Editor

Major Revisions

Your manuscript has been reviewed by two expert reviewers and they have provided various suggestions for additions and improvements. I agree with Reviewer 2 (Gary Rosenberg) that your experimental design could be improved and hope you will able to incorporate these items in your revision. He also raises some interesting points about the status of Clessina stelzneri striata, as you will see.

Reviewer 1 ·

Basic reporting

This paper is original and constitutes an important and valuable progress in the knowledge of the endemic gastropods from Argentina. It discusses the taxonomy of genus Clessinia using multiple sources: shell and anatomical morphology, shell morphometry and molecular data. A molecular phylogeny is presented and the authors discuss a previous hypothesis of paraphyly for Clessinia and Spixia.
The manuscript is clearly written in professional English and well organized. The title is accurate and the abstract summarises correctly the study´s content. The introduction and literature cited account correctly the background information. Figures and tables are clear and relevant to the content of the article. All data provided as supplemental files

Experimental design

The paper consists of original primary research.
The research hypotheses are clear and relevant to clarify the taxonomy of genus Clessinia.
The investigation was conducted rigorously, the methods are reproducible and adequate to test the hypotheses proposed.
The two morphometric analyses performed (lineal and geometric) showed similar results. Please consider eliminate one of them to reduce the length of the manuscript.
A phylogeny based on morphology and a discussion of evolution of characters and consistencies/inconsistences with the molecular phylogeny would be desirable. Please explain why you decided not to perform this kind of analysis.

Validity of the findings

The multiple-source data provided in the paper to test hypotheses of species delimitation in genus Clessinia are accurate and support the conclusions.

Additional comments

Please, consider the following suggestions to improve the article:
-Line 178. Material and methods: Please include the number of specimens (shells and living ones) used for anatomical descriptions and molecular analyses
- Line 207: 10 specimens from each species or subspecies is a small sample size to perform a morphometric analysis. The variability within and among species can be underestimated and some overlap could not be detected. Please, consider redoing the analysis with a higher N of each taxon.
- Regarding geometric morphometry, as landmarks are considered homologous morphological loci, please explain how you could determine homologous landmarks in whorl sutures in snails with different whorl number. (In case, you consider only snails with the same number of whorls, some variability is missing)
-Line 342, 343, 351. I suggest using only “lamellae” here and employing “teeth” in the context of digestive system (radula).
-Line 657. Please include Remarks to compare the subspecies C. stelzneri stelzneri and C. stelzneri striata
-Line 1076 and the following ones. Discussion of the coat of sand granules in haired shells.
Line 1090. Authors stated: “Strikingly dryer habitats hold the species with longer hairs…” Please clarify why this fact is strikingly. May be, the authors could discuss Pfenninger et al. (2005) [Why do snails have hairs? A Bayesian inference of character evolution].
The authors explained correctly that ornamentation of shells could be adaptive (anti-predatory/reproductive behavior) or passive (a by-product of the passage through a sticky substrate or by the adherence to ribs, hairs or spines on the shell). The adaptive explanations must be tested through a series of experiments to demonstrate heritability and variability of the character, correlation between the character and fitness different from 0, etc. Therefore, adaptive explanations in the current state of knowledge are speculative. I suggest not mentioning the hypothesis related to anti-predatory behavior in the Abstract (line 60) and Conclusions (lines 1145 and 1146).
-Figure 2. Legend: species names should be in italics
- Figure 5. Legend: C. cordovana and C. pagoda should be in italics
- Figure 7. Legend: C. stelzneri striata should be in italics
- Figure 8. Legend: C. stelzneri striata should be in italics
- Figure 10. Legend: C. pagoda should be in italics
- Figures 12 and 15. Legend: Please specify that limits penis/epiphallus and epiphallus/flagellum are indicated. It would be helpful to indicate these limits also in figures 6 and 9.
-Figure 16. The axes are described in the text as Discriminant function1 and 2 and in Figure 16 as Canonical axe 1 and 2. Please change the axis names in Figure 16.

·

Basic reporting

Overall summary: You have generated interesting data sets for shell, periostracum, reproductive anatomy and DNA sequences toward your stated objective “to revise the species of the genus Clessinia, under a morphological, geometric morphometrics and molecular combined approach”. Three taxa are subject to morphometric analysis and seven to molecular analysis with 16S and COI (twelve with additional sequences from GenBank for COI.)

Basic reporting

The paper does not follow a standard structure in that the results start with“Systematic accounts”, yet the classification adopted should be a conclusion of the paper, not a result. The placement of the Systematic accounts gives the appearance that the names used for the taxa were decided a priori rather than being determined by the analyses. I recommend that the Systematic accounts be moved to an Appendix. The description of the ecoregions within it (starting at line 363) might be moved to the introduction (or results if the information in relation to snails is novel). The information on “Taxonomic history of the genus” (lines 127 to 170, but not 171-175) in the introduction should also be moved the Appendix. This information is important for getting the priority of names of taxonomic entities right, but it distracts from the main focus of the paper: determining what entities should be recognized.

You give explicit results for Morphometrics and for Sequence Data, but not for periostracal or anatomical characters. The abstract emphasizes that “the importance of the periostracal ornamentation for species identification needed to be tested”, so stating these results clearly is necessary. Periostracum and reproductive anatomy are described in the Systematic accounts, but a summary of the findings is not presented until the Discussion. Both character sets distinguish striata from stelzneri: “in C. stelzneri striata the periostracal hairs are absent and the spiral rows are more scatter” (lines 997-998). and “the penis in C. cordovana and C. stelzneri striata shows a short, thin, penial sheath overlapping part of the penial distal portion. This sheath was not observed in C. stelzneri stelzneri” (lines 1006-1008).


The literature cited and background information are thorough, except that you do not address what species concept(s) you are using, and what criteria you use to distinguish species from subspecies. The distribution map in Figure 2 shows that the geographic ranges of stelzneri and striata overlap, so under a standard geographic subspecies concept, the rank of subspecies for striata is disproven. Either C. striata is a synonym of C. stelzneri, or it is a full species (or the distribution map is incorrect).

Background information on Odontostomidae is given in the first two paragraphs of the Discussion. This should be moved into the Introduction. The discussion also contains a considerable amount of information on the periostracum (lines 1062 to 1101) some of which could be moved to the Introduction.

Another area in which background information could be improved for the general reader is clearly definition of terms relating to Chaco. You use “Dry Chaco” as the broadest term for the ecoregion, with “Arid Chaco”, “Semiarid Chaco”, and “Chaco Serrano” (lines 383-384) as sub-ecoregions. In the caption to Fig. 2, “Arid Chaco” is apparently given as “Chaco”. Since some sources use “Chaco” and “Arid Chaco” as two separate ecoregions (e.g. https://www.worldwildlife.org/ecoregions/nt0701), this is confusing. “Espinal” and “Yungas” are also distinct ecoregions, but that isn’t stated in the figure caption. The colors are explained in Fig 2A. If the dark brown in Fig. 2B refers to the same region as in Fig. 2A, the key to colors should be at the level of the figure, not the subfigure.

The results heading of the abstract states “We found that species of Clessinia are endemic to the Chaco Serrano sub-ecoregion that is restricted to small patches within the Dry Chaco area. If that is actually a result of the paper, rather than background information, it needs to be brought out more clearly. Also, it needs evaluation to see if it is an overgeneralization. Figure 2 shows one site for striata is in Semiarid Chaco (west to Totoral) rather than Chaco Serrano, and some sites for each of the other taxa in “Chaco”, not Chaco Serrano.

In general the writing is unambiguous, but there is some non-standard English, for example use of “posterior” to mean “subsequent”. I provide some suggestions under “General comments” below.

Experimental design

The general idea of the study is sound, taking several lines of evidence and combining them to see what set of taxa is best supported. However, the methods do not provide a decision criterion—how to determine what taxa to recognize when the data sets contradict each other. In this case periostracum, anatomy, morphometrics all support more species than does ABGD. I recommend as a starting point Schlick-Steiner et al. (2010 Integrative Taxonomy: A multisource approach to exploring biodiversity. Annual Review of Entomology 55:421-438 (https://doi.org/10.1146/annurev-ento-112408-085432).

You state (line 274) that “Phylogenetic analyses were performed using Neighbour-Joining (NJ), and Bayesian inference (BI)”. Neighbor joining is a phenetic method, not a phylogenetic method. It doesn’t incorporate a model of evolution, so it is not appropriate for this kind of study. Apparently it gave the same topology as the Bayesian analysis, since bootstrap values for NJ are mapped onto the Bayesian tree, but this is not stated. It would be more appropriate to do parsimony or likelihood analysis as alternatives to Bayesian in exploring the sequence data. See this discussion on ResearchGate https://www.researchgate.net/post/Neighbor_joining_or_maximum_likelihood

Only mitochondrial genes were used for the phylogenetic analysis. A nuclear gene should be added as an independent marker. Mitochondrial genes may not yield phylogenies consistent with species relationships because of introgression. There is evidence that this has occurred in Clessinia. In the phylogenetic trees of Breure & Romero (2012), whom you cite, C. cordovana and C. stelzneri grouped with Spixia popana in the phylogram (fig. 7) based on the mitochondrial gene COI, but with only nuclear genes, they grouped with S. tucamanensis instead (figs. 5-6). Your results are more similar to their figure 7, showing the need to add a nuclear gene to the study.

Another problem with the phylogenetic analysis is that only a single outgroup was used. This makes the outgroup a long branch; including several outgroups would give a better assessment of polarity of characters.

Barcode distances were calculated based on only two or three specimens per species, but 10 per species is usually recommended (e.g., https://www.ncbi.nlm.nih.gov/pmc/articles/PMC2775153/). There may not have been enough data for a reliable determination of a barcode gap since there were few intraspecific comparisons and each putative species came from only one or two locations. Also,it is not clear how the histogram in Fig. 20A was compiled. It shows only 2 genetic distance values between 0.01 and 0.02, but Table 4 show 8 such values. The stated genetic distances between C. stelzneri stelzneri and C. stelzneri striata “1.1 to 1.8%” (lines 1014-1015) are too low; this is the distance for 16S (Table 3), but for COI, the barcode gene, distances are 1.3 to 3.9% (Table 4; note that C. s. stelzneri are IDs 16, 17 and 21 in this table). The higher end of the range is certainly consistent with species rank in many barcoding studies.

For the morphometric part of the study you state that the points “are homologous in all individuals in the analysis” (line 230), however, many of the landmarks on the spire are not true homologues. If individuals have different numbers of whorls, then you might be comparing, for example, whorl 6 to whorl 7.

It appears that points were determined differently between the discriminant analysis and the canonical variate analysis. In the former, height of aperture (Hap, Fig. 1E) was measured from the top of the aperture to the point on the lip directly below it on the shell axis. Diameter of aperture (Dap) was measured orthogonal to Hap, from the left most point of the inner lip to the intercept with the outer lip. However, these values are better determined by measuring between tangent lines parallel or perpendicular to the shell axis, as was done with minor diameter (dm, fig. 1D). The points to use for the tangents are shown in Fig. 1G, LM 10 to LM 8 for Hap and LM9 to LM7 for Dap.

In Fig 1C, diameter of spire (DS) is shown measured from whorl to suture instead of whorl to whorl. In Fig, 1A, the lines showing how the measurements were made are displaced from the figure, down and to the right.

Given that both of the morphometric analyses did fully separate the 3 taxa involved, these points about measurement may be of minor concern. However, all of the measured snails appear to have come from single populations (judging from the first table in supplemental data) in which case some of the differences could have been ecophenotypic. Broader geographic inclusion of individuals in the morphometric analysis would give a stronger assessment if the putative taxa are morphometrically distinct.

Validity of the findings

You don’t make it clear why only three species are subject to morphometric analysis, whereas twelve are in the molecular analysis. I think this is because the current taxonomic position of both Clessinia stelzneri and striata is as subspecies of Clessinia cordovana, which is not stated in the manuscript. Parodiz’s striata has never been treated as a subspecies of stelzneri (see chresonymy at line 574); Clessinia stelzneri striata is actually new combination introduced in the manuscript. Also, it is not the case as stated on lines 504 and 987 that Breure (1974) elevated stelzneri to full species status. He classified it on pages 110 and 113 as C. cordovana stelzneri. His mention of C. stelzneri cordovana (p. 123) is an error. His listing of Clessinia stelzneri on p. 124 is intended to refer to the original description, and does not indicate his own taxonomic placement. Breure & Romero (2012) treated the taxon as C. cordovana stelzneri. It seems unlikely that Neubert & Janssen, 2004 (line 504) used stelzneri as a full species, as their work is a type catalogue. This leaves Cuezzo et al. (2013), who did rank it as a full species, but this appears to be a misinterpretation of Breure (1974).

This all works to your benefit. As it is, the manuscript appears merely to verify the status quo, confirming two species and one subspecies. But actually, you have confirmed that stelzneri should be ranked as a full species as first named, and proposed that striata be a subspecies of stelzneri rather than of cordovana. I do not think you should take the latter action, however, because striata has geographic overlap with stelzneri, which precludes subspecies rank. Instead I think you should raise striata to full species status.

Your morphological and morphometric data sets each support full species status for all of the entities they include. Your conclusions, however, consider only the ABGD analysis, which is based solely on barcode gap, to suggest that C. s. stelzneri, C. s. striata and Spixia popana are conspecific. I do not think the results of the ABGD analysis are supported by the weight of your data. A more thorough analysis in the context of integrative taxonomy is needed, and I provide some suggestions toward that end below.

Puillandre et al. (2012), whom you cite, state “ABGD is fast, simple method to split a sequence alignment data set into candidate species that should be complemented with other evidence in an integrative taxonomic approach.” Give that all of your other data sets support full species status for striata, and the small sample size and lack of nuclear genes cast doubt on the ABGD, it seems that full species status for all the taxa considered should be the preferred conclusion. Low genetic distance between of some species might reflect recent speciation, or be a symptom of some of the methodological problems indicated.

Your data support the synonymy of Spixia Pilsbry & Vanatta with Clessinia Doering, 1875. If that is correct, Clessinia striata (Parodiz, 1939), whether ranked as a subspecies or a species, is preoccupied by Pupa striata Spix, 1827, the type species of Spixia, and needs a new name. this gives you the opportunity to name a new species with a new type series, a clear type locality, and a DNA sequenced holotype, rather than merely introduce a replacement name for striata.

The suggestion that the periostracal hairs function in camouflage should be identified as speculation. Another potential function could be to attract moisture. You might also consider in explaining periostracal function, why striata lacks periostracal hairs.

Additional comments

Title: remove the author and year of the genus. They are cited in the body of the paper so they don’t need to be in the title, since the paper is not about the date of publication. It might be worth adding “Argentina” to the title.

Line 31: change “conform” to “comprise”.

Line 32: change “scarcely” which is a bit colloquial, to “rarely”.

Line 34 and 88: say “rare endemic” rather than “endemic rare”.

Line 86: delete “Even,”.

Line 92: “during the last decade” This cites a study from 2009, which would seem then to refer to two decades ago, not the immediately past decade.

Line 99: “Doering 1874 [1875]”. This is a confusing form of reference. In the running text and references cited, the correct date under the International Code of Zoological Nomenclature should be given (1875). Under the heading for the taxon, more detail can be given on the dates of publication if needed. The stated versus true dates of publication can also be stated in brackets after the entry in the References Cited. This applies to references to “Doering 1875 [1877]”.

Line 103-105: Change “The distribution area of Clessinia is highly overlapped with species of Spixia Pilsbry & Vanatta, 1898, and in less proportion with Plagiodontes” to “The distribution area of Clessinia is overlaps largely that of Spixia Pilsbry & Vanatta, 1898, and to a lesser degree with that of Plagiodontes.”

Line 107: change “the doubt” to “doubt”.

Line 123-124: delete “on the basis of our new molecular study”; new evidence is already referenced at the start of the sentence.

Line 141: change ‘who read it from a handwritten label wrote” to “based on a label written”

Line 152: change “which name-bearing type is” to “typified by”.

Line 158-159 “following years”. Grammatically this means years after Breure & Miquel (2012), but the following paragraph refers to earlier events.

Line 160: change “Posterior” to “Later” or “Subsequent”.

Line 179: state where voucher specimens for the morphological and molecular studies are deposited, IBN and/or IFML (from line 301), and their catalogue numbers.

Line 180-184 “Hand collection…”. This should have its own heading, “Collecting and Preservation” rather than being under “Morphological Studies”, since the work also supported the molecular studies.

Line 181: snails rather than areas are “xerophilic”.

Line 183-184: “Fixation”, “fixed”. Alcohol is a preservative not a fixative.

Line 183: “alcohol 96%”, alcohol 75%; say “96% alcohol” and “75% alcohol; also state what kind of alcohol, presumably ethanol (mentioned on line 247).

Line 185: delete “and studying them”.

Line 186: change “where” to “were”.

Line 201: change “plain” to “plane”.

Line 203-24: “The number of whorls was calculated following Kerney & Cameron (1979)”. Number of whorls shouldn’t be under the heading for linear morphometrics as it wasn’t used in that analysis (Table 2).

Line 205-206: “Measurements of type material of each species, is recorded in the species description…” This also doesn’t belong under the heading for linear morphmetrics.

Line 206: change “minimum” to “minimum”.

Line 232: change “digitalized” to “digitized”.

Line 298: change “punctual” to “point”.

Line 328: Technically what is labeled here as “Diagnosis” is a “Definition”; see the glossary of the ICZN. The diagnosis is the direct comparison to other taxa that follows (line 336-354).

Line 328: change “turriteliform” to turritelliform”.

Line 349: change “Type” to “type”.

Line 351: change “teeth less” to “toothless”.

Line 506: also list Breure & Romero (2012) for this combination.

Line 857 and 881: change “discriminate” to “discriminated”.

Line 906: change “none allometric” to “non-allometric”.

Line 912: change “with the detached of aperture respect to body whorl” to “to detachment of the aperture”.

Line 919: change “all two” to “both”.

Line 922-923: “Spixia cuezzoae clustered with specimens of Clessinia pagoda”—so did Spixia holmbergi.

Line 924-925: change “COI- based dataset” to “larger COI dataset including sequences from GenBank.”

Line 926: change “well supported” to “well-supported”.

Line 929: “tri-modal”. Fig. 20A doesn’t have enough structure to say it is tri-modal.

Line 940: change “Pulmonate” to “pulmonate”.

Line 963: change “differencing” to “differentiating”.

Line 978-979: “Nevertheless, morphometric information alone did not add precise information respect to the assignment of all specimens”. This is not true, the discriminant analysis achieved 100% success (line 887).

Line 985: change “Posteriorly” to “Subsequently”

Line 985-986: “C. stelzneri was elevated again to species (Breure, 1974).” No, Breure 1974 said “status: C. cordovana stelzneri)”, p. 110 and 113. Breure & Romero (2012) also treated stelzneri as a subspecies of C. cordovana.

Line 1102: change “By last” to “Finally” or delete it.

Line 1103: delete ‘Based on morphological evidence, current”.

Line 1104-1105, change ‘a detached shell body whorl aperture” to “the aperture detached from the body whorl”.

Line 1116: change “from” to “to”.

Line 1117: change “formed by” to “contains”.

Line 1118-1119: change “only known molecular sequences for four species (Breure & Romero, 2012), plus two new other added in the present study.” to “DNA sequences are known for only six species, including two added by the present study”.

Supplementary data
Linear measurements:

1) Worksheet labelled C. cordovana has data on three taxa, not only C. cordovana.
2) In Column A, delete “VENTRAL” since both lateral and ventral data are given.
3) Delete extra worksheets

Field work permision in Catamarca
This was submitted as a “rar” file, which I don’t have the software to open. I recommend it be converted to something more standard, like pdf.

“Lateral links” and “Ventral links for geometric morphometrics”
The purpose of these files is not stated; a description should be added.

GenBank numbers
This file can be deleted since GenBank numbers are given in Table 1 in the manuscript.

---

## Round 0.2 · Minor Revisions

· Academic Editor

Minor Revisions

As you will see, your manuscript has been re-reviewed by Dr. Rosenberg who has a number of mostly small suggestions. In my view, the most important one is the suggested clarification concerning your new name (i.e., please make explicit that this is not a replacement name). :

·

Basic reporting

The authors have revised the manuscript extensively in response to reviewers comments, improving the logical flow, and making the background information clearer, for example, refining the discussion of ecoregions and redrafting map figures so that terms are used in a consistent way through the manuscript and distributions are consistent with the terminology.

Experimental design

The authors in their rebuttal show that they have carefully considered their experimental design. They have not accept all of the reviewers’ suggestions, for example stating, “Although we used a multiple source of information we do not support the Integrative Taxonomy approach”. They cite appropriate literature in rebuttal to show that they are familiar with the debate. They have improved the experimental design by adding genetic data from a nuclear gene and morphometric data on individuals from more populations. As a result, they give a more balanced evaluation of their data, and have revised some of their conclusions.

Validity of the findings

The most significant change is recognition of all the taxa at species rank, rather than ranking one as a subspecies. That taxon, Clessinia tulembensis sp. nov. (as Clessinia stelzneri striata in the first submission) will probably be the main point of controversy in the manuscript. I agree with the authors that the taxon deserves species rank, but I think they can go farther in defending their decision.

Clessinia tulumbensis is genetically quite similar to C. stelzneri, and the phylogenetic analysis shows C. stelzneri as paraphyletic with respect to C. tulumbensis (Figures 3 and 4), which is not noted in the discussion. Paraphyly is not problematic--species unlike higher taxa are not required to be monophyletic in a phylogenetic classification—but it does show something interesting about the direction of evolution. Also, the authors describe the morphological differences between these taxa, but do not state that they have overlapping distributions as show in fig. 5D. These species are apparently sympatric, if not syntopic, without morphological intergrades. That should be part of the argument for giving full species status to C. tulumbensis. A similar case of snails with strong morphological difference but little genetic difference because of recent divergence was published by Uit de Weerd and Velázquez (2017).

Uit de Weerd, D. R. and A. F. Velázquez. 2017. Pinning down Tenuistemma (Pulmonata: Urocoptidae): local evolution of an extreme shell type. Biological Journal of the Linnean Society 121: 741–752, doi 10.1093/biolinnean/blx041.

Additional comments

Custom Checks requested by PeerJ
• DNA data: The authors have an appropriate statement of deposition of data in GenBank and cite GenBank numbers for sequences by taxon in Table 2. The data cannot yet be accessed on GenBank, presumably because the authors have requested it not be released until publication of the corresponding manuscript.
• Field study: The authors present appropriate permits for their work in supplementary documents.
• New species: I agree that the taxon they describe is a new species, and the authors have included an the statement required statement in the manuscript [As an ICZN commissioner, I have to say that this statement is unnecessary; give the ZooBank registration number to meet ICZN requirements. No explanation is needed

I am concerned, however, to ensure that the authors do not run afoul of ICZN Article 72.7:
“If an author proposes a new species-group name expressly as a replacement (a nomen novum) for an earlier available one, then the two names are objective synonyms; both the nominal taxa they denote have the same name-bearing type despite any simultaneous restriction or application of the new replacement name (nomen novum) to particular specimens or any contrary designation of type, or any different taxonomic usage of the new replacement name.”

The authors are describing a new species to include the taxon named Odontostomus cordovanus striatus Parodiz, 1939. That name is preoccupied by Pupa striata Spix, 1827. They do not intend to introduce a new replacement name, but rather to name a new species with its own type series, so that the species is defined based on live-collected material from which DNA sequences were obtained and the anatomy described. That is, the intent is that Parodiz’s taxon and their taxon will be subjective synonyms, not objective synonyms.

The authors state, however, “The former Clessinia cordovana striata (Parodiz, 1939) is here raised to species rank, renaming it as Clessinia tulumbensis sp. nov.” It is possible that someone could invoke ICZN Article 72.7 to say that there is a contrary designation of types. This challenge could fail because the authors say “sp. nov.” rather than “nom. nov.” but it seems better to prevent the challenge from occurring to begin with. The authors should say that although the Parodiz name is preoccupied, they are not replacing it, but naming a new species for the reasons noted above.

General comments
Most of the following are just suggestions to improve phrasing.
Line 56: change “suggest” to “suggests”.
Line 112 change “is overlaps” to “overlaps” [I think this was my error from the first review].
Line 116, change “stressed” to “exacerbated”.
Line 195 change “is” to “are” to agree with “limits”.
Line 199 change “stablished” to “established”.
Line 413: change “by any of both methods” to “by either method”.
Line 440: self reference “Cuezzo, Miranda, Vogler and Beltramino, 2018” isn’t needed; Spixia should be attributed to its original publication here along with citation any works that used Spixia in that sense. “New Synonymy” should be lower case.
Line 457: change “teeths” to “teeth”.
Line 465: “ranging its area of distribution to”—not sure what is meant here. Possibly replacing this phrase with “in” would work.
Line 479; delete comma.
Line 484: change ‘of extension” to “in extent”.
Line 513 and 584: delete the dash, these are citations of original descriptions.
Line 518: also mention type material of C. gracilis.
Line 578: change “with” to “as”.
Line 627: change “idem to” to “same as in”.
Line 644: delete “The distribution area of C. stelzneri is located in”.
Line 650: delete dash before Parodiz.
Line 653: also mention type material of O. cordovana striata since it is not an objective synonym.
Line 813-814: Recommend deleting “C. pagoda is a rare species, endemic to a restricted area in northwestern Córdoba.” This distribution information does not contribute anything relative to the first sentence, and “rare” contradicts the following sentence, which says “easily found”, which suggests the taxon can be locally common.
Line 819: change “adheres” to “adhere”.
Line 823: delete “immerse”.
Line 848: change “rows” to row”. It’s not clear what is meant by, “Only one type of spiral rows present, some of them without spines intercalated with the ones bearing spines.” If some have spines and some don’t isn’t that two types of spiral rows?
Line 891: change ‘floor” to “ground”?
Line 902: change “the genera/species” to “taxonomic”.
Line 903: change “provides” to “provide” and “its” to “their”.
Line 906: change “this” to “the”.
Line 916: Periostracum doesn’t need definition here as it’s already been used dozens of times.
Line 927: change “dryer” to “drier”.
953-997: break this into two paragraphs.
Line 972: change “The same than” to “As”.
Fig. 2: a different green is used to the text for C. stelzneri than for the symbol.
Fig. 17B. The B is hard to see, maybe change to white?

---

## Round 0.3 · accepted · Accept

· Academic Editor

Accept

Thank you for addressing the remaining points of the "minor revisions." I am happy to accept your manuscript at this time.

#